# Evaluation of Monte Carlo tools for high energy atmospheric physics II : relativistic runaway electron avalanches

David Sarria[1], Casper Rutjes[2], Gabriel Diniz[2,3], Alejandro Luque[4], Kevin M. A. Ihaddadene[5], Joseph R. Dwyer[5], Nikolai Østgaard[1], Alexander B. Skeltved[1], Ivan S. Ferreira[3], and Ute Ebert[2,6]

[1]Birkeland Centre for Space Science, Department of Physics and Technology, University of Bergen, Bergen, Norway
[2]Centrum Wiskunde & Informatica (CWI), Amsterdam, The Netherlands
[3]Instituto de Física, Universidade de Brasília, Brazil
[4]Instituto de Astrofísica de Andalucía (IAA-CSIC), P.O. Box 3004, Granada, Spain
[5]University of New Hampshire Main Campus, Department of Physics, Durham, NH, United States
[6]Eindhoven University of Technology, Eindhoven, The Netherlands

*Correspondence to:* David Sarria (david.sarria@uib.no)

**Abstract.**

The emerging field of High Energy Atmospheric Physics studies how high energy particles are produced in thunderstorms, in the form of terrestrial gamma-ray flashes and gamma-ray glows (also referred as thunderstorm ground enhancements). Understanding these phenomena requires appropriate models of the interaction of electrons, positrons and photons with air molecules and electric fields. We investigated the results of three codes used in the community (Geant4, GRRR and REAM) to simulate Relativistic Runaway Electron Avalanches (RREAs). This work continues the study of Rutjes et al. (2016), now also including the effects of electric fields, up to the classical breakdown field, which is about 3.0 MV/m at standard temperature and pressure.

We first present our theoretical description of the RREA process, that is based and incremented over previous published works. This analysis confirmed that the avalanche is mainly driven by electric fields and the ionisation and scattering processes determining the minimum energy of electrons that can runaway, that was found to be above $\approx 10$ keV for any fields up to the classical breakdown field.

To investigate this point further, we then evaluated the probability to produce a RREA as a function of the initial electron energy and of the magnitude of the electric field. We found that the stepping methodology in the particle simulation has to be set up very carefully in Geant4. For example, a too large step size can lead to an avalanche probability reduced by a factor of 10, or to a 40% over-estimation of the average electron energy. When properly set-up, both Geant4 models show an overall good agreement (within $\approx 10$ %) with REAM and GRRR. Furthermore, the probability that particles below 10 keV accelerate and participate in the high energy radiation is found negligible for electric fields below the classical breakdown value. The added value of accurately tracking low energy particles ($< 10$ keV) is minor, and mainly visible for fields above 2 MV/m.

In a second simulation set-up, we compared the physical characteristics of the avalanches produced by the four models: avalanche (time and length) scales, convergence time to a self-similar state and energy spectra of photons and electrons. The two Geant4 models and REAM showed a good agreement on all parameters we tested. GRRR was also found to be consistent

with the other codes, except for the electron energy spectra. That is probably because GRRR does not include straggling for the radiative and ionisation energy losses, hence implementing these two processes is of primary importance to produce accurate RREA spectra. Including precise modelling of the interactions of particles below 10 keV (e.g. by taking into account molecular binding energy of secondary electrons for impact ionisation) also produced only small differences in the recorded spectra.

## 1 Introduction

### 1.1 Phenomena and observations in high energy atmospheric physics

In 1925, C.T.R. Wilson proposed that thunderstorms could emit a "measurable amount of extremely penetrating radiation of $\beta$ or $\gamma$ type" (Wilson, 1925), about 60 years before such radiation was observed from the atmosphere and from space (Parks et al., 1981; Fishman et al., 1994; Williams, 2010). This, and subsequent observations and modeling are now being investigated withing the field of High Energy Atmospheric Physics (HEAP). A review is provided by Dwyer et al. (2012).

Observationally different types of high energy emissions have been identified coming from thunderclouds, naturally categorized by duration. Microsecond-long burst of photons, which were first observed from space (Fishman et al., 1994; Grefenstette et al., 2009; Marisaldi et al., 2014; Roberts et al., 2018), are known as Terrestrial Gamma-ray Flashes (TGFs). TGFs also produce bursts of electron and positrons (Dwyer et al., 2008; Briggs et al., 2011; Sarria et al., 2016) that follow the geomagnetic field lines into space and show longer durations. Two space missions specifically designed to study TGFs and related phenomena will provide new observations in the near future : ASIM (Atmosphere-Space Interaction Monitor) (Neubert et al., 2006), successfully launched in April 2018; and TARANIS (Tool for the Analysis of Radiation from lightning and Sprites) (Lefeuvre et al., 2009; Sarria et al., 2017) to be launched at the end of 2019.

Seconds to minutes or even hours long X and gamma radiation have been observed on ground, from balloons and aircraft, by McCarthy and Parks (1985); Eack et al. (1996); Torii et al. (2002); Tsuchiya et al. (2007); Adachi et al. (2008); Chilingarian et al. (2010, 2011); Kelley et al. (2015); Dwyer et al. (2015); Kochkin et al. (2017, 2018), which are called gamma-ray glows or thunderstorm ground enhancements. Some modeling attempts of both gamma ray and electron observations are also presented in Chilingarian et al. (2012).

TGFs were predicted to create a neutron emission on the millisecond duration, with associated isotope production (Babich, 2006). Such emission was observed from the ground (Bowers et al., 2017; Teruaki et al., 2017). A similar phenomenon was modeled at higher altitudes by Rutjes et al. (2017), that also proposed to call it "TGF afterglow".

Following the idea of Wilson (1925), high energy X and gamma radiation is created by runaway electrons, which may further grow by the effect of Møller scattering in the form of so called relativistic runaway electron avalanches (RREAs) (Gurevich et al., 1992). For the multiplication to occurre, a threshold electric field of $E_{\mathrm{th}} = 0.28$ MV/m (at STP) is required (Babich et al., 2004a; Dwyer, 2003).

The difference in duration between TGFs and gamma-ray glows can be explained by two possible scenarios to create runaway electrons, which is traditionally illustrated using the average energy-loss or friction curve (see, e.g., figure 1 of Dwyer et al. (2012)). In this curve, there is a maximum at around $\varepsilon \approx 123$ eV, illustrating the scenario that for electric fields higher

than a critical electric field, of $E_c \approx 26$ MV/m at standard temperature and pressure (STP), thermal electrons can be accelerated into runaway regime, described in the so-called Cold Runaway theory (Gurevich, 1961). The effective value of $E_c$ may be significantly lower, as electrons could overcome the friction barrier due to their intrinsic random interactions (Lehtinen et al., 1999; Li et al., 2009; Liu et al., 2016; Chanrion et al., 2016). Cold Runaway could happen in the streamer phase (Moss et al., 2006; Li et al., 2009; Chanrion and Neubert, 2010) or leader phase (Celestin and Pasko, 2011; Celestin et al., 2012; Chanrion et al., 2014; Köhn et al., 2014; Köhn and Ebert, 2015; Köhn et al., 2017) of a transient discharge, explaining the high energy electron seeding that will evolve to RREAs and produce gamma-rays by bremsstrahlung emission from the accelerated electrons. The cold runaway mechanism may be further investigated with laboratory experiments, in high voltage and pulsed plasma technology, and may be linked to the not fully understood x-ray emissions that have been observed during nanosecond pulsed discharges and the formation of long sparks, (Rahman et al. (2008); Dwyer et al. (2008); Shao et al. (2011); Kochkin et al. (2016), and references therein), with different possible production mechanism that were proposed and tested using analytical modeling (Cooray et al., 2009) and computer simulations (Ihaddadene and Celestin, 2015; Luque, 2017; Lehtinen and Østgaard, 2018). Alternatively, the relativistic feedback discharge model is also proposed to explain TGF production using large scale and high potential electric fields (Dwyer, 2012), where the RREA initial seeding may be provided by cosmic-ray secondaries, background radiation, or cold runaway (Dwyer, 2008).

For fields significantly below the thermal runaway critical electric field $E_c \approx 26$ MV/m but above the RREA threshold electric field of $E_{\text{th}} = 0.28$ MV/m (at STP), runaway behaviour is still observed in detailed Monte Carlo studies (see Dwyer et al. (2012), and references therein). At thundercloud altitudes, cosmic particles create energetic electrons that could runaway in patches of the thundercloud where the electric field satisfies this criterion. RREAs are then formed if space permits and could be sustained with feedback of photons and positrons creating new avalanches (Babich et al., 2005; Dwyer, 2007, 2012). Gamma-ray glows could be explained by this mechanism, as they are observed irrespectively of lightning or observed to be terminated by lightning (McCarthy and Parks, 1985; Chilingarian et al., 2015; Kelley et al., 2015; Kochkin et al., 2017). The fact that gamma-ray glows are not (necessarily) accompanied by classical discharges, results in the conclusion that the electric fields causing them are usually also below the conventional breakdown. The conventional (or classical) breakdown field, of $E_k \approx 3.0$ MV/m (at STP), is where low energy electrons ($< 123$ eV) exponentially grow in number, as ionisation overcomes attachment. This exponential growth of charged particles will affect the electric field, which requires a self-consistent simulation to be properly taken into account. That is not something we want to test in this study, since Geant4 is not capable of simulating it. Therefore we will focus on electric fields below the breakdown field $E_k \approx 3.0$ MV/m, and above the RREA threshold $E_{\text{th}} \approx 0.28$ MV/m.

As a note, one can find in the literature that $E_k$ can be given between 2.36 MV/m and 3.2 MV/m (Raizer, 1997), the theoretical lowest breakdown field being between 2.36 and 2.6 MV/m (see Raizer, 1997, page 338). The value of $\approx 3.2$ MV/m is the measured breakdown field in centimeter gaps in laboratory spark experiments (see Raizer, 1997, page 135), that can be lower for longer gaps.

## 1.2 Theoretical understanding of RREAs

In the energy regime of a kilo-electronvolt (keV) to a hundred of mega-electronvolts (MeV), the evolution of electrons is mostly driven by electron impact ionisation (Landau et al., 2013), as this energy loss channel is much larger than the radiative (bremsstrahlung) energy loss. However, the bremsstrahlung process does impact the shape of the electron energy spectrum, that can be understood by the straggling effect, that is discussed in the next section. When the electric field is below the classical breakdown $E_k \approx 3.0$ MV/m (at STP), the system can be simplified, because the effect of the electrons below a certain energy can be neglected, in particular the population that would otherwise (if $E > E_k$) multiply exponentially and have an important effect on the electric field. The part of the electron population that decelerates, and eventually attaches, cannot contribute to the production of the high energy radiation. Let $\epsilon_2^{\min}$ be the minimum energy for a secondary electron to have a chance to runaway, thus participate to the production of high energy radiation. The subscript index $i = 2$ indicates a secondary electron. A precise value of $\epsilon_2^{\min}$ will be evaluated in section 3 with the help of simulations, but, by looking at the friction curve, one can guess it is located in the keV to tens of keV energy regime (see Dwyer et al., 2012, Figure 1). As almost all energy loss of ionisation is going into producing secondary electrons of lower energy ($\epsilon_2 \lesssim 200$ eV), it is reasonable to approximate that channel as a continuous energy loss, or friction.

In the case of electric fields above the RREA threshold ($E_{\mathrm{th}} = 0.28$ MV/m at STP), the electrons, when considered as a population, will undergo avalanche multiplication. Some individual electrons do not survive (because there can be hard bremsstrahlung or ionisation collisions that will remove enough energy to get below $\epsilon_2^{\min}$), but the ensemble grows exponentially as new electrons keep being generated from the ionisation collisions on air molecules, including a fraction with energy larger than $\epsilon_2^{\min}$. The production of secondaries with energies much larger than the ionisation threshold (a few kilo-electronvolts being a reasonable value), can be described using the Møller cross section, which is the exact solution for a free-free electron-electron interaction (see, e.g., Landau et al. (2013, page 321)) :

$$\frac{\mathrm{d}\sigma_{\mathrm{M}}}{\mathrm{d}\delta_2} = Z\frac{2\pi r_e^2}{\gamma_1^2 - 1}\left[\frac{(\gamma_1 - 1)^2\gamma_1^2}{\delta_2^2(\gamma_1 - 1 - \delta_2)^2}\right.$$
$$\left. - \frac{2\gamma_1^2 + 2\gamma_1 - 1}{\delta_2(\gamma_1 - 1 - \delta_2)} + 1\right], \tag{1}$$

where $Z$ is the number of electrons in the molecule, the index $i = 1$ indicates the primary electron, $i = 2$ the secondary, $\gamma_i$ is the Lorentz factor, $\delta_i = \gamma_i - 1 = \epsilon_i/(m_e c^2)$ is the kinetic energy divided by the electron rest energy (with rest mass $m_e$) and $r_e = \frac{1}{4\pi\epsilon_0}\frac{e^2}{m_e c^2} \approx 2.8 \times 10^{-15}$ m is the classical electron radius. In the case $\delta_2 \ll \gamma_1 - 1$ and $\delta_2 \ll 1$, we observe that the term $\propto 1/\delta_2^2$ is dominating. Thus, we can write equation 1 as:

$$\frac{\mathrm{d}\sigma_{\mathrm{M}}}{\mathrm{d}\delta_2} \approx Z\frac{2\pi r_e^2}{\beta_1^2}\frac{1}{\delta_2^2}, \tag{2}$$

with $\beta_1 = v_1/c$ the velocity of the primary particle. Integrating equation (2) from $\delta_2$ to the maximum energy ($\epsilon_1/2$) yields a production rate

$$\sigma_{\mathrm{prod}} \approx Z\frac{2\pi r_e^2}{\beta_1^2}\frac{1}{\delta_2} \propto \frac{1}{\epsilon_2}, \tag{3}$$

using again $\epsilon_2 \ll \epsilon_1$. The remaining sensitivity of $\sigma_{\mathrm{prod}}$ in units of area to the primary particle is given by the the factor $\beta_1^2$ which converges strongly to 1 as the mean energy of the primary electrons exceeds 1 MeV. In other words, as the mean energy of the electrons grows towards even more relativistic energies, the production rate $\sigma_{\mathrm{prod}}$ becomes independent of the energy spectrum.

For illustrative purposes, we now consider the one dimensional deterministic case, which results in an analytical solution of the electron energy spectrum. We make the system deterministic by assuming that the differential cross section is a delta-function at $\epsilon_2^{\min}$ (the minimum energy at which a secondary electron can runaway) and use $\Lambda_{\mathrm{prod}} = \frac{1}{N\sigma_{\mathrm{prod}}}$ as the constant collision length, with $N$ the air number density. In other words, every length $\Lambda_{\mathrm{prod}}$ a secondary electron of energy $\epsilon_2^{\min}$ is produced. The derivation below is close to what was presented by Celestin and Pasko (2010); Dwyer et al. (2012); Skeltved et al. (2014) and references therein.

Consider a population of electrons in one dimension with space-coordinate $z$, a homogeneous and constant electric field $E$ above the RREA threshold and a friction force $F(\epsilon)$. The minimum energy $\epsilon_2^{\min}$ at which an electron can runaway is given by the requirement $F(\epsilon_2^{\min}) \approx qE$ (where $q$ is the elementary charge), that is to say $\epsilon_2^{\min} = \mathrm{function}(F, E)$ is constant. Assuming that the mean energy of the ensemble is relativistic results in a constant production rate $\Lambda_{\mathrm{prod}} = \Lambda_{\mathrm{prod}}(\epsilon_{\min})$. Thus, in space, the distribution $f_e$ grows exponentially as,

$$\frac{\partial f_e}{\partial z} = \frac{1}{\Lambda_{\mathrm{prod}}} f_e. \tag{4}$$

While in energy, the differential equation is given by the net force,

$$\frac{\mathrm{d}\epsilon}{\mathrm{d}z} = qE - F(\epsilon). \tag{5}$$

Solving for steady state means,

$$\frac{\mathrm{d}f_e}{\mathrm{d}z} = \frac{\partial f_e}{\partial z} + \frac{\partial f_e}{\partial \epsilon}\frac{\mathrm{d}\epsilon}{\mathrm{d}z} = 0, \tag{6}$$

and using equation 4 and 5 results in,

$$\frac{\partial f_e}{\partial \epsilon} = -\frac{1}{\Lambda_{\mathrm{prod}}(qE - F(\epsilon))} f_e. \tag{7}$$

For the largest part of the energy spectrum, specifically above 0.511 MeV and below 100 MeV, $F(\epsilon)$ is not sensitive to $\epsilon$ (e.g. see Rutjes et al. (2016)). Only at around $\epsilon \approx 100$ MeV electron energy $F(\epsilon)$ starts increasing again because of the bremsstrahlung process. Thus, one may assume $F(\epsilon) \approx F$ constant, which yields that the RREA energy spectrum $f(\epsilon)$ at steady state is given by,

$$f_e(\epsilon) = \frac{1}{\bar{\epsilon}} \exp\left(-\frac{\epsilon}{\bar{\epsilon}}\right), \tag{8}$$

with the exponential shape parameter and approximated average energy $\bar{\epsilon}(E)$ given by,

$$\bar{\epsilon}(E) = \Lambda_{\mathrm{prod}}(qE - F). \tag{9}$$

Equivalently, in terms of collision frequency $\nu_{\text{prod}} = \frac{\beta c}{\Lambda_{\text{prod}}}$, equation 9 can be written as,

$$\bar{\epsilon}(E) = \frac{\beta c}{\nu_{\text{prod}}}(qE - F), \qquad (10)$$

with $\beta$ the velocity $v/c$ of the RREA avalanche front. For the 1-d case there is no momentum-loss or diffusion, so $\beta \approx 1$. Note that $\Lambda_{\text{prod}}$ depends on $\epsilon_2^{\min}$ (the minimum energy at which a secondary electron can runaway), which depends on the electric field $E$ as that determines the minimum electron energy that can go into runaway. In this analysis, we illustrate with equations 8 and 9, that the full RREA characteristics, such as the mean energy $\bar{\epsilon}$ or the collision length $\Lambda_{\text{prod}}$ (directly related to the avalanche length scale $\lambda$ discussed in section 4.1) are driven by processes determining $\epsilon_2^{\min}$.

In reality there are important differences compared to the one dimensional deterministic case described previously, which we propose to discuss qualitatively for understanding the Monte Carlo simulations evaluated in this study. During collisions, electrons deviate from the path parallel to $E$. Therefore in general, electrons experience a reduced net electric field as the cosine function of the opening angle $\theta$, which reduces the net force to $qE\cos(\theta) - F$ and thereby the mean energy $\bar{\epsilon}$ of equation 9. In reality the 3D scattering (with angle parameter $\theta$) changes of the path of the particle. Although the velocity remains still close to $c$ (as the mean energy is still larger than several MeV), the RREA front velocity parallel to the electric field ($\mathbf{E}$) is reduced again because of the opening angle as function of its cosine:

$$\beta_{\parallel} = \beta\cos(\theta), \qquad (11)$$

which also reduces the mean energy $\bar{\epsilon}$. Note that $\theta$ is not a constant and may change with each collision. Equivalently the avalanche scale length $\Lambda_{\text{prod}}$ in 3-D changes to $\approx \Lambda_{\text{prod}} \times \cos(\theta)$. However, most importantly, the momentum-loss of the lower energetic electrons results in a significant increase of $\epsilon_2^{\min}$, as it is much harder for electrons to runaway. The increase of $\epsilon_2^{\min}$ significantly increases $\Lambda_{\text{prod}}$ and thereby increases the characteristic mean energy $\bar{\epsilon}$. On the other hand, the stochasticity creates an interval of possible energies $\epsilon_2^{\min}$ that can runaway with a certain probability and for thin targets a straggling effect (Rutjes et al., 2016). A recent article discussed the influence of the angular scattering of electrons on the runaway threshold in air (Chanrion et al., 2016).

The effects discussed above prevent a straight forward analytical derivation of the RREA characteristics in 3 dimensions, but what remains is the important notion that the physics is completely driven by the intermediate energy electron production. "Intermediate" means they are far above ionisation threshold ($\gg 123$ eV) but much below relativistic energies ($\ll 1$ MeV). The parametrisation of the electron energy spectrum, given by equation 9 turns out to be an accurate empirical fit, as it was already shown in Celestin and Pasko (2010); Dwyer et al. (2012); Skeltved et al. (2014) and references therein. Nevertheless in these works $\lambda^{\min}(E)$, or equivalently the velocity over collision frequency $\beta c/\nu_{\text{prod}}$, is fitted by numerical Monte Carlo studies and the final direct relation to $\epsilon_2^{\min}$ is not executed. Celestin and Pasko (2010) calculated that $\nu^{\text{prod}}(E) \propto E$, thus explains why $\bar{\epsilon}(E)$ must saturate to constant value. Celestin and Pasko (2010) argue that $\epsilon_2^{\min}(E)$ is given by the deterministic friction curve $F$, for which they use the Bethe's formula and an integration of a more sophisticated electron impact ionisation cross section (RBEB) including molecular effects, but that is only true in one dimension without stochastic fluctuations. Other attempts to simulate RREA by solving the kinetic equation instead of using Monte-Carlo methods are presented in Roussel-Dupre et al.

(1994); Gurevich and Zybin (1998); Babich et al. (2001) and references therein. An analytical approach is provided by Cramer et al. (2014).

## 1.3   Model reductions and previous study

Apart from analytical calculations, the physics behind TGFs, TGF afterglows and gamma-ray glows are also studied with the help of experimental data, computer simulations, and often a combination of both. Simulations necessarily involves model reduction and assumptions. As we argued in the previously, in scenarios where the electric field is below the classical breakdown field ($E_k \approx 3.0$ MV/m at STP), electrons below a certain energy can be neglected, because they will decelerate and eventually attach, thus not contributing to the production of the hard radiation. In Monte Carlo simulations it is therefore common to apply a so-called "low energy cutoff" (or threshold), noted $\varepsilon_c$, that is a threshold where particles with lower energy can be discarded (or not produced), to improve code performance. It is different from $\epsilon_2^{\min}$ (the minimum energy at which a secondary electron can runaway) as one is a simulation parameter and the other is a physical value. Ideally, $\varepsilon_c$ should be set as close as possible to $\epsilon_2^{\min}$. A second simplification can be made for the energetic enough particles that stay in the ensemble, by treating collisions that would produce particles below the low energy cutoff as a friction.

Both simplifications can be implemented in different ways, leading to different efficiencies and accuracies. Rutjes et al. (2016) benchmarked the performance of the Monte Carlo codes Geant4 (Agostinelli et al., 2003), EGS5 (Hirayama et al., 2005), FLUKA (Ferrari et al., 2005) developed in other fields of physics, and of the custom-made codes GRRR (Luque, 2014) and MC-PEPTITA (Sarria et al., 2015) within the parameter regime relevant for HEAP, in the absence of electric and magnetic fields. In that study they focused on basic tests of electrons, positrons and photons with kinetic energies between 100 keV and 40 MeV through homogeneous air using a low energy cutoff of 50 keV and found several differences between the codes and invited other researchers to test their codes on the provided test configurations. We found that the usage of an average friction fails in the high energy regime ($\gtrsim 100$ keV), as the energy loss is too much averaged, resulting in an incorrect energy distribution (Rutjes et al., 2016).

As we indicated in section 1.2, the ionisation energy loss channel is much larger than the radiative (bremsstrahlung) energy loss, by a few orders of magnitude. However, this is only true for the average, and bremsstrahlung does have a significant effect on the electron spectrum because of straggling (Rutjes et al., 2016). This straggling effect was first studied by Bethe and Heitler (1934). If it is not taken into account in the implementation of the low energy cut-off, the primary particle suffers a uniform (and deterministic) energy loss. This means that only the energy of the primary particle is altered, but not its direction. The accuracy of the assumed uniform energy loss is a matter of length scale : on a small length scale, the real energy loss distribution (if all interactions are considered explicitly) among the population would have a large spread. One way to obtain an accurate energy distribution is by implementing a stochastic friction mimicking the straggling effect.

Rutjes et al. (2016) also indicated that including electric fields in the simulations would potentially enhance the differences found by introducing new errors, the simulation results being supposingly sensitive to the low energy cutoff. This effect is believed to be responsible of the observed differences between the two Geant4 physics lists tested in Skeltved et al. (2014): for all fields between 0.4 and 2.5 MV/m (at STP), they found that the energy the spectrum and the mean energy of runaway

electrons depended on the low energy cutoff, even when it was chosen between 250 eV and 1 keV. In the following, this interpretation is challenged.

## 1.4 Content and order of the present study

In the context of High Energy Atmospheric Physics, the computer codes that were used are either general purpose codes developed by large collaborations, or custom made codes programmed by smaller groups or individuals. Examples of general purpose codes that were used are Geant4 (e.g., Østgaard et al., 2008; Carlson et al., 2010; Bowers et al., 2017; Sarria et al., 2015, 2017; Skeltved et al., 2014) and FLUKA (e.g., Dubinova et al., 2015; Rutjes et al., 2017). Custom made codes were used in Roussel-Dupre et al. (1994); Lehtinen et al. (1999); Dwyer (2003); Babich et al. (2004b); Østgaard et al. (2008); Celestin and Pasko (2011); Luque (2014); Köhn et al. (2014); Chanrion et al. (2014); Sarria et al. (2015), among others. Rutjes et al. (2016) presented in their section 1.3 the reasons why different results between codes (or models) can be obtained and why defining a comparison standard (based on the physical outputs produced by the codes) is the easiest way (if not the only) to compare and verify the codes. Here we continue the work of Rutjes et al. (2016), now with electric fields up to the classical breakdown field ($E_k \approx 3.0$ MV/m). As mentioned previously, we chose not to use larger electric fields because that would produce an exponential growth of low energy electrons ($< 123$ eV) which would affect the electric field and therefore require a self-consistent simulation, that Geant4 is not capable of. We aim to provide a comparison standard for the particle codes able to simulate Relativistic Runaway Electron Avalanches, as simple and informative as possible, by only considering their physical outputs. These comparison standards are described in the Supplementary Material (Sections 1 and 2).

In section 1.2, we illustrated that the full RREA characteristics, such as the mean energy $\bar{\epsilon}$ or the collision length $\Lambda_{\text{prod}}$ are driven by processes determining $\epsilon_2^{\text{min}}$ (the minimum energy at which a secondary electron can runaway). To prove this insight, and to benchmark codes capable of computing RREA characteristics for further use, we calculated the probability for an electron to accelerate into the runaway regime (see section 3), which is closely related to the quantity $\epsilon_2^{\text{min}}$. From this probability study, it is directly clear that it is safe to choose the low energy cutoff $\varepsilon_c$ higher than previously expected by Skeltved et al. (2014) and Rutjes et al. (2016), given an electric field $E < E_k$. In section 3, we will demonstrate that the probability for particles below 10 keV to accelerate and participate in the penetrating radiation is actually negligible. Thus, in practice an energy threshold value of $\varepsilon_c \approx 10$ keV can be used for any electric field below $E_k$. However, in section 2.4, we will show that step-length restrictions are of major importance (e.g. it can lead to an underestimation of a factor of 10 of the probability to produce a RREA, in some cases). The results of the comparison of several parameters of the RREAs produced by the four tested codes is then presented in section 4. We conclude in section 5.

The test set-ups of the two types of simulations (RREA probability, and RREA characteristics) are described in the Supplementary Material, together with the data we generated, and supplementary figures comparing several characteristics of the showers. The Geant4 source codes used in this study are also provided (see sections 6 and 7).

## 2 Model descriptions

The data we discuss in the next sections were produced by the general-purpose code Geant4 (with several set-ups) and two custom-made codes (GRRR and REAM) which we describe below. However, we will not describe comprehensively all the processes, models or cross-sections used by the different codes, and a table summarizing it is provided in the supplementary material document, section 13 (including all references).

### 2.1 Geant4

Geant4 is a software toolkit developed by the European Organization for Nuclear Research (CERN) and a worldwide collaboration (Agostinelli et al., 2003; Allison et al., 2006, 2016). We use version 10.2.3. The electro-magnetic models can simulate the propagation of photons, electrons and positrons including all the relevant processes, and the effect of electric and magnetic fields. Geant4 uses steps in distance, whereas REAM and GRRR use time step. In the context of this study, three main different electro-magnetic cross-section sets implementation are included : one based on analytical of semi-analytical models (e.g. uses the Møller cross section for ionisation and Klein-Nishina cross section for Compton scattering), one based on the Livermore data set (Perkins et al., 1991), and one based on the Penelope models (Salvat et al., 2011). Each of them can be implemented with a large number of different electro-magnetic parameters (binning of the cross section tables, energy thresholds, production cuts, maximum energies, multiple scattering factors, accuracy of the electro-magnetic field stepper, among others), and some processes have multiple models in addition to the main three, e.g. the Monash University model for Compton scattering (Brown et al., 2014). Skeltved et al. (2014) used two different physics list : LHEP and LBE. The first one, based on parametrisation on measurement data and optimized for speed, was deprecated since the 10.0 version of the toolkit. The LBE physics list is based on the Livermore data, but it is not considered as the most accurate electro-magnetic physics list in the Geant4 documentation, which is given by the Option 4 physics list (O4). This last uses a mix of different models, and in particular uses the Penelope model for the the impact ionisation of electrons. For this study, we will use two GEANT4 physics list options : Option 4 (referred as simply O4 hereafter) that is the most accurate one according to the documentation, and the Option 1 (referred as simply O1 hereafter) that is less accurate, but runs faster. In practice, O1 and O4 give very similar results for simulations without electric field and energies above 50 keV, as produced in our previous code comparison study (Rutjes et al., 2016).

By default, Geant4 is following all primary particles down to zero energy. A primary particle is defined as a particle with more energy than a threshold energy $\varepsilon_c^g$ (that is different from $\varepsilon_c$ described before). By default, $\varepsilon_c^g$ is set to 990 eV and was not changed to obtain the results presented in the next sections. The LBE Physics list used by Skeltved et al. (2014) uses a threshold down to 250 eV (i.e. more accurate than using 990 eV, in principle) and this parameter was thought to be responsible for a major change in the accuracy of the obtained RREA energy spectra. In section 3, we will argue that the most important factor able to effect the spectra obtained from Geant4 simulations is the accuracy of the stepping method for the tracking of the electrons, and not the low energy threshold. Actually, we found that the stepping accuracy of the simulation is indirectly improved by reducing $\varepsilon_c^g$, that explains why Skeltved et al. (2014) could make this interpretation.

## 2.2 GRRR

The GRRR (GRanada Relativistic Runaway simulator) is a time-oriented code for the simulation of energetic electrons propagating in air, and can handle self-consistent electric fields. It is described in detail in the supplement of Luque (2014) and its source code is fully available in a public repository (see section 7 about code availability). In the scope of this work, we want to point out three important features : 1. Electron ionisation and scattering processes are simulated discretely, and the friction is uniform and without a way to mimic the straggling effect. 2. Bremsstrahlung collisions are not explicit and are simulated as continuous radiative losses, without straggling. 3. GRRR uses a constant time-step $\Delta t$ both for the integration of the continuous interactions using a fourth-order Runge-Kutta scheme and for determining the collision probability of each discrete process $k$ as $\nu_k \Delta t$, where $\nu_k$ is the collision rate of process $k$. This expression assumes that $\nu_k \Delta t \ll 1$ and therefore that the probability of a particle experiencing two collisions within $\Delta t$ is negligibly small. The collisions are sampled at the beginning of each time step and therefore the rate $\nu_k$ is calculated using the energy at that instant. In this work we used $\Delta t = 0.25$ ps for the avalanche probabilities simulations, and $\Delta t = 1$ ps for the simulations used to characterise the RREA. For both cases, the time steps are small enough to guarantee a very accurate integration.

## 2.3 REAM

The REAM (Runaway Electron Avalanche Model) is a three dimension Monte Carlo simulation of Relativistic Runaway Electron Avalanche (also refereed as Runaway Breakdown), including electric and magnetic fields (Dwyer, 2003, 2007; Cramer et al., 2016). This code is inspired by earlier work by Lehtinen et al. (1999) and takes accurately into account all the important interactions involving runaway electrons, including energy losses through ionisation, atomic excitation and Møller scattering. A shielded-Coulomb potential is implemented in order to fully model elastic scattering, and it also includes the production of X/gamma-rays from radiation energy loss (bremsstrahlung) and the propagation of the photons, by including photoelectric absorption, Compton scattering and electron/positron pair production. The positron propagation is also simulated, including the generation of energetic seed electrons through Bhabha scattering. The bremsstrahlung photon emissions from the newly produced electrons and positron are also included.

In the scope of this study, it is important to point out that REAM limits the time step size of the particles so that the energy change within one time step cannot be more than 10 %. The effect of reducing this factor down to 1 % was tested and did not make any noticeable difference in the resulting spectra. The comparative curves are presented in the Supplementary Material, section 10.

## 2.4 Stepping methodology

### 2.4.1 General method

In Monte Carlo simulations, particles propagate in steps, collide and interact with surrounding media by means of cross sections (and their derivatives). A step is defined by the displacement of a particle between two collisions. As it is presented in

sections 3 and 4, the stepping methodology is responsible for most of the differences we observed between the codes we tested. Simulations can be either *space-oriented* or *time-oriented*, if the stepping is done in space or in time. By construction, space-oriented simulations are thus not synchronous in time. Usually, a single particle is simulated until it goes below the low energy threshold ($\varepsilon_c$), chosen by the user. But there are exceptions, like Geant4, that by default follows all primary particles down to zero energy. The advantage of asynchronous simulations is the ability to easily include boundaries, to have particles step as far as possible in the same material (minimizing the overhead due to null collisions), and smaller memory usage since there is no need to store all the particles alive at a given time (that may be a million or more). However, asynchronous simulations makes it impossible to incorporate particle to particle interactions, such as a space charge electric field, or self-consistent electric fields.

During steps, charged particles can lose energy (and momentum) by collisions, and also change in energy (and momentum) when an electric fields is present. To guarantee accuracy, energies should be updated frequently enough. An accurate method would be to exponentially sample step lengths with

$$\delta\ell = \min_{\epsilon}\{(\sigma_t(\epsilon)N)^{-1}\}, \tag{12}$$

in space-oriented perspective, or

$$\delta t = \min_{\epsilon}\{(v(\epsilon)\sigma_t(\epsilon)N)^{-1}\}, \tag{13}$$

in time-oriented perspective. With $v$ the velocity, $\sigma_t$ the total cross section and $N$ the number density of the medium. Then at each updated location (and energy), the type of collision must be sampled from probability distributions. The probability of doing a collision of the given process ($pr$) can be calculated with:

$$p_{pr} = 1 - \exp\left(-N\int_i^f \sigma_{pr}\left(\epsilon(\ell)\right)\,d\ell\right) \tag{14}$$

Where the index $i$ refers to the beginning of the step, $f$ to its end, $\ell$ is the step length variable along the trajectory, and $d\ell$ is an infinitesimal step length. For time oriented simulations, we have equivalently :

$$p_{pr} = 1 - \exp\left(-N\int_i^f v(\epsilon(t))\,\sigma_{pr}\left(\epsilon(t)\right)\,dt\right) \tag{15}$$

Using these probabilities along a given step length or duration, there is a chance that no interactions happens, but the energy of the particle is guaranteed to be updated correctly.

### 2.4.2 The case of Geant4

In the Geant4 documentation, the stepping method presented in the previous section is referred as the "the integral approach to particle transport". This method is set up by default in Geant4 for impact ionisation and bremsstrahlung. However, the way it is

implemented is not exactly following what was described in the previous section. The description of the exact implementation is out of the scope of this article, but is presented into details in Ivanchenko et al. (1991) and Apostolakis et al. (2009). The method relies on determining the maximum of the cross section over the step ($\sigma_{max}$), using a parameter $\alpha_R$ (called "dR over Range" in the Geant4 documentation), that is also used to determine the step lengths. Another related parameter is the maximum range parameter ($\rho_{max}$), set to the default values of 1 millimeter and 0.1 millimeter for O1 and O4 respectively, and was never changed in the scope of this study. The exact definition of these parameters is given in Allison et al. (2016) and in the online Geant4 documentation (available at http://geant4.cern.ch/support/userdocuments.shtml). The default value of $\alpha_R$ is set to 0.80 for O1 and to 0.20 for O4. We found that both values are not low enough to be able to produce accurate results for the RREA probability simulations presented in the next section. To make Geant4 able to produce accurate RREA simulations using the multiple scattering algorithm, two methods are possible.

The first method is to tweak the value of the $\alpha_R$ parameter. Its value is set to 0.80 by default for O1, and 0.20 by default for O4. We found that these default values are way too high to be able to produce accurate RREA simulations, and values of $\alpha_R < 5.0 \times 10^{-3}$ should be used, as presented in the next section.

The second method is to implement a step limiter process (or maximum acceptable step). By default, this max step ($\delta\ell_{max}$) is set to one kilometer, and such a large value has no effect in practice, since the mean free path of energetic electrons in STP air is orders of magnitude smaller. Acceptable values of $\delta\ell_{max}$ depend on the electric field, and we found that it should be set to 1 millimeter or less to produce accurate RREA simulations, as presented in the next section. However, using this method results in relatively long simulation time required to achieve an acceptable accuracy, as the step is not adapted to the energy of the electrons. For information, the relative impact on performance (in terms of requirements of computation time) of tweaking the $\delta\ell_{max}$ and $\alpha_R$ parameters is presented in Appendix A.

## 3   Probability of generating RREA

As a first comparison test, we estimated the probability for an electron to accelerate into the runaway regime and produce a RREA, given its initial energy $\epsilon$ and some electric field magnitude $E$. Note that the momentum of the initial electrons is aligned along the opposite direction of the electric field, so that it gets accelerated. That gives maximum RREA probabilities, as other alignments reduce the chance to produce a shower (see, e.g., (Lehtinen et al., 1999, Figure 2.6)). We defined this probability as the fraction of initial (seed) electrons that created an avalanche of at least 20 electrons above 1 MeV. Once this state is reached, there is no doubt the RREA is triggered and can go on forever if no limits are set. The number 20 is arbitrary, to be well above 1 but small enough for computational reasons. For some initial conditions, we also tested requirements of 30 and 50 electrons above 1 MeV, that resulted in very similar probabilities. This study is somehow similar to the works presented in Lehtinen et al. (1999); Li et al. (2009); Liu et al. (2016); Chanrion et al. (2016), but they all looked at the probability to have only one single runaway electron, whereas we used the criterion of $N = 20$ electrons above 1 MeV, that is a stricter constraint. The difference between the two criteria is mainly noticeable for low electric field ($< 0.4$ MV/m) and high seed energies ($> 700$ keV). A figure illustrating how the probability can change with $N$ is presented in the supplementary material, section 5.3.

As a test case, we calculated the probability to produce RREAs as a function of $\alpha_R$ and $\delta\ell_{max}$ (these parameters are presented in the previous section), for the configuration $\epsilon = 75$ keV, $E = 0.80$ MV/m. This case was chosen because it showed a particularly large sensitivity to the stepping methodology, as discussed later. The results are presented in figure 1. Although this configuration has a very low RREA probability for O1 and O4 by default (where $\alpha_R$ respectively equal to 0.80 and 0.20, and $\delta\ell_{max}$ is one kilometer for both), the probability increases as $\alpha_R$ decreases and converges to a value between 10 and 12 % for both models when $\alpha_R < 5.0 \times 10^{-3}$. The same effect is observed when reducing $\delta\ell_{max}$. In this case, the user should not set $\delta\ell_{max}$ below the maximum range parameter, set to 1 millimeter for O1 and 0.1 millimeter for O4 by default (and never changed in the scope of this article). When reducing the $\alpha_R$ parameter to arbitrarily small values, both Geant4 models converge to slightly different probabilities : 10.7 % for O1 and 11.7 % for O4. We think this small difference is not due to the stepping method, as reducing $\rho_{max}$ or $\alpha_R$ further does not produce a significant difference. It is probability due to other factors, in particular the difference in the physical models and cross section sets used.

As explained in section 1.2, the final electron spectrum is essentially driven by the minimum energy $\epsilon_2^{\min}$ of electrons that can create a RREA. Here we can clearly see this probability is strongly affected by the choice of the $\alpha_R$ and $\delta\ell_{max}$ simulation parameters, affecting the accuracy of the stepping method, and that the values set by default for these parameters are not precise enough to obtain correct RREA probabilities. In order to help future researchers, we provide example Geant4 source codes where the $\alpha_R$ and $\delta\ell_{max}$ parameters can be changed and their effect to be tested (see sections 6 and 7).

In figure 2.a, we compare the contour lines of the 10%, 50% and 90% probability of triggering a RREA as function $\epsilon$ and $E$, for the four models : Geant4 O4 ($\alpha_R = 1.0 \times 10^{-3}$), Geant4 O1 ($\alpha_R = 1.0 \times 10^{-3}$), GRRR and REAM. The full RREA probability results in the $(\epsilon, E)$ domain for each model are presented in the supplementary material, section 5.

The most important difference between Geant4 and GRRR is present for energies $> 200$ keV and E-fields $< 0.5$ MV/m. At 1 MeV, the level curves are significantly different between the Geant4 models and GRRR: the 50% probability to trigger RREA for GRRR is approximately located at the 10 % probability for O4, and the 90 % probability to for GRRR is located at the 50 % probability for O1. The reason is probably similar to a point we raised in our previous study (Rutjes et al., 2016) : GRRR does not include a way to simulate the straggling effect for the ionisation process. By looking at figure 2 of Rutjes et al. (2016), we can see that 200 keV is roughly the energy from where the difference in the spectrum of GRRR, compared to codes that simulate straggling, starts to become significant.

For low electron energy ($< 40$ keV) and high electric field ($> 2$ MV/m), GRRR and O4 present a good agreement, however O1 deviates significantly from O4. We investigated the effects of the stepping parameters ($\alpha_R$, $\delta\ell_{max}$ and $\rho_{max}$) and it is clear that they were not involved in this case. We think the Møller differential cross section (with respect to the energy of the secondary electron) used by O1 and extrapolated down to low energies leads to the production of secondary electrons with average energies lower than the Penelope model (used by O4), that includes the effects of the atomic electron shells, hence is probably more accurate. This hypothesis is confirmed by looking at the shape of the differential cross sections of impact ionisation, which plots are presented in the Supplementary Material, section 11.4.

The RREA probability data for REAM is also displayed in figure 2.a, as the red curves. The three REAM level curves show a significantly higher noise than the Geant4 data, mainly because the latter used 1000 electrons seeds whereas the former used

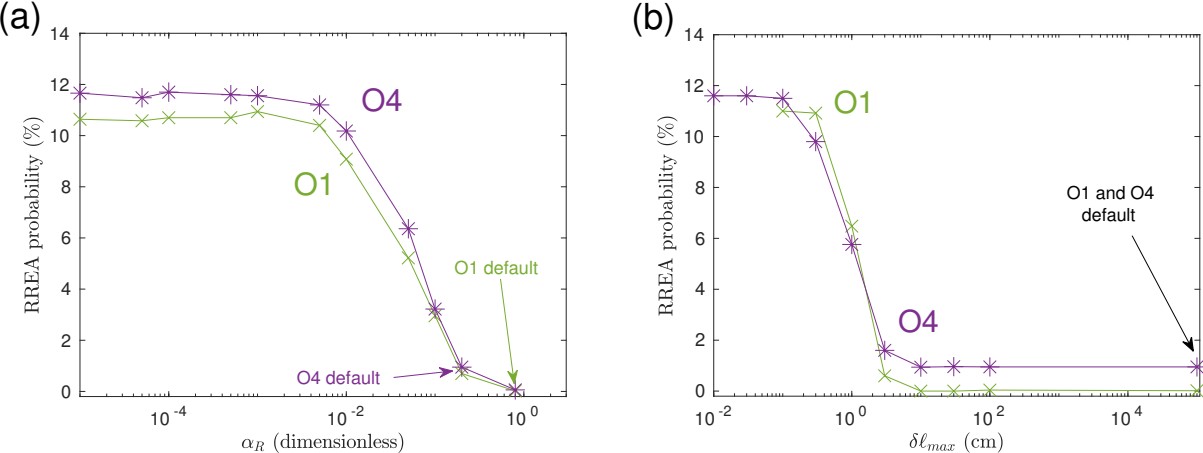

**Figure 1.** Relativistic avalanches probabilities calculated from Geant4 simulations, for specific point $\{\epsilon = 75 \text{ keV}, E = 0.80 \text{ MV/m}\}$ (illustrated by a cross in figure 2) and for two stepping settings. **(a)**: Avalanche probability versus $\alpha_R$ setting for Geant4 O4 and Geant4 O1. $\delta\ell_{max}$ is set to the default value of 1 kilometer. **(b)**: Avalanche probability versus maximum step setting ($\delta\ell_{max}$) for Geant4 O4 and Geant4 O1. The parameter $\alpha_R$ is set to the default value of the models, that is 0.8 for O1 and 0.2 for O4.

only 100. The algorithms used to calculate the levels curves were also found to impact the noise level. Nevertheless, the noise level is low enough to be able to evaluate the consistency between the codes. REAM shows a consistency with Geant4 (O1 and O4) within less than 12% in the full parameter range, and less than 5 % in some part of it. The most apparent deviations between REAM and Geant4 O1/O4 can be noticed for a seed electron energy range between 50 and 300 keV, for the 50 %

and 90% level curves, where there is a systematic, statistically significant difference in the probability for REAM compared to Geant4 (REAM requiring about 10% larger electric field or primary electron energy to reach the 90% or the 50% contour level). However, we do not expect such a small difference to significantly affect the characteristics of the RREA showers, such as the multiplication factors or the mean energies of the RREA electrons. To test this quantitatively, a detailed comparison of the most important characteristics of the RREA showers obtained with the four models is presented in the following section.

In figure 2.b we show the 0 %, 10 %, 50 %, 90 % and 100 % probability contour lines for the Geant4 O4 model where we could run a very large number of initial electrons ("seeds") to obtain curves with a very low noise level. These are the most accurate probabilities we could obtain. From this figure, it is clear that the RREA probability for an electron of less than $\approx 10$ keV is null for any electric field below $E_k \approx 3.0$ MV/m. Therefore 10 keV is a reasonable a lower boundary of $\epsilon_2^{\min}$ (the minimum energy at which a secondary electron can runaway), and any simulation with an electric field below $E_k \approx 3.0$ MV/m

could use an energy threshold ($\varepsilon_c$) of this value while keeping accurate results. If electric fields with lower magnitude are used, it is also reasonable to increase this energy threshold by following the 0% level curve showed in Figure 2.b.

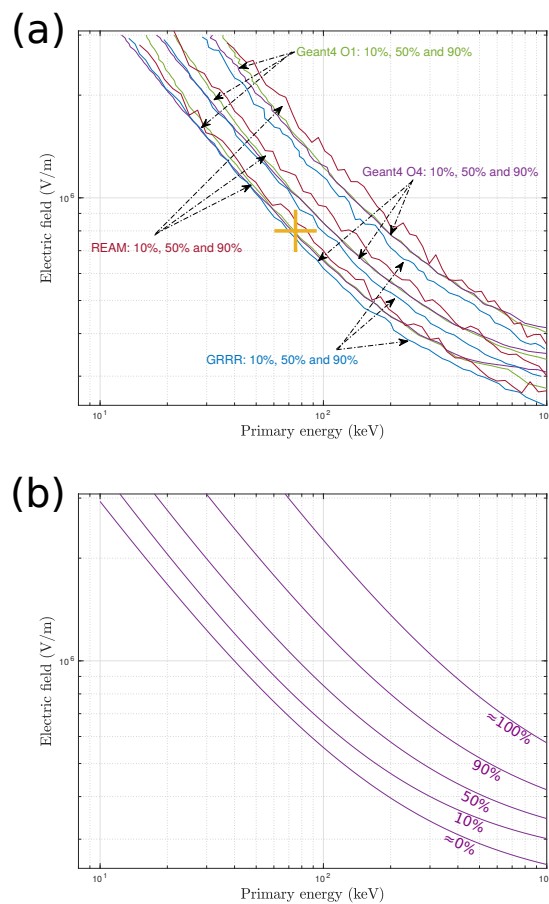

**Figure 2. (a)**: Relativistic avalanche probability comparison between GRRR, REAM, O4 and O1. It shows three contour lines at 10%, 50% and 90%, as function of seed (primary) energy $\epsilon$ and electric field magnitude $E$. These contours are derived from the full probability scan, that are presented in the Supplementary Material (section 5). The cross at $\{\epsilon = 75 \text{ keV}, E = 0.80 \text{ MV/m}\}$ highlights the point where we studied the effect of the simulation stepping parameters (for the O4 and O1) on the probability, see figure 1. **(b)**: Five contour lines indicating the 0%, 10%, 50%, 90% and 100% probabilities to generate a relativistic electron avalanche (RREA) as function of $\epsilon$ and $E$, for the Geant4 O4 model for which we could run a very large number of initial electrons ($> 50,000$) to obtain curves with a very low noise level.

## 4 Characteristisation of RREA showers

We compared the output of the four models over 12 different electric field magnitudes from $E = 0.60$ MV/m to $E = 3.0$ MV/m. Two types of simulation were set : record in time, and record in distance (or space). This last choice was made because the resulting spectra can change significantly depending on the record method, as presented in figure 10 of Skeltved et al. (2014). All the curves presenting the simulation results are presented in the Supplementary Material, as well as the complete details on how the simulation should be set up. In the following section, we discuss only the most important differences we found between the four codes. We show the comparison of avalanche scales in space and time in section 4.1 and in section 4.2 the evolution to self-similar state. Finally, in section 4.3 we show the comparison of the self-similar energy spectra of electrons and photons of the RREA.

### 4.1 Avalanche time and length scales

Figures 3 and 4 show the avalanche length and time scales as function of electric fields, for the four models, together with their relative difference with respect to REAM. Note that we could not compute any values for electric fields below 0.60 MV/m, as we only used 200 initial electron seeds of 100 keV, which could not produced enough showers. The choice of 200 initial electrons is purely due to computational limitations. The avalanches length and times of the different models agree within $\pm 10\%$. There is also a systematic shift of about 7 % between the two Geant4 models for both time and length scales. The Geant4 O4 model is in principle more accurate than the O1 model, since it includes more advanced models. For most of the electric fields, O1 tends to be closer to REAM and O4 tends to be closer to GRRR. Following Coleman and Dwyer (2006), the avalanche length and time can be fitted by the empirical models,

$$\lambda(E) = \frac{c_1}{E - c_2}, \tag{16}$$

$$\tau(E) = \frac{c_3}{E - c_4} \tag{17}$$

where $c_1$ is in V, $c_2$ and $c_4$ in V/m and and $c_3$ in s ·V/m. The $c_2$ and $c_4$ parameters can be seen as two estimates of the magnitude of the electric field of the minimum of ionisation for electrons along the avalanche direction, and also of the electric field magnitude of the RREA threshold; both values being close. However, we note that these fits neglect the sensitivity of the mean energy and velocity to the electric field. These empirical fits are motivated from the relations presented in equation 9 and 10, derived for the one dimensional case. First results of such fits were presented in Babich et al. (2004b) and Coleman and Dwyer (2006); and they obtained consistent results. Here we will compare our results against Coleman and Dwyer.

The best fit values of the two models to the simulation data are given in table 1. The $c_1$ parameter is directly linked to the average energy of the RREA spectrum, though the definition of this average energy can be ambiguous as energy spectra change significantly if recorded in time or in space. The values given by all the code are located between 6.8 and 7.61 MV, and

**Table 1.** Values of the parameters of the fits (with 95 % confidence intervals) for the simulations data for avalanche scale in space and time, using the models described by equations 16 and 17. See figure 3 and 4 for the corresponding curves.

| Code | Avalanche length | | Avalanche time | |
|------|-------------------|--|----------------|--|
|      | $c_1$ (MV) | $c_2$ (kV/m) | $c_3$ (ns MV/m) | $c_4$ (kV/m) |
| REAM | $7.43 \pm 0.18$ | $290 \pm 9.5$ | $27.6 \pm 0.91$ | $293 \pm 13$ |
| G4 O1 | $7.50 \pm 0.10$ | $276 \pm 5.6$ | $27.6 \pm 0.44$ | $290 \pm 6.3$ |
| G4 O4 | $6.93 \pm 0.13$ | $285 \pm 7.5$ | $25.9 \pm 0.28$ | $288 \pm 4.2$ |
| GRRR | $7.25 \pm 0.30$ | $266 \pm 18$ | $26.2 \pm 0.76$ | $282 \pm 12$ |

are all consistent with each other within a 95 % confidence interval, with the exception of O4 that slightly deviates from O1. Combining the four values gives :

$$\overline{c_1} = 7.28 \pm 0.10 \text{ MV} \tag{18}$$

By "combining", we mean that the four values are averaged and the rule $\sigma_{comb} = \sqrt{\sigma_1^2 + \sigma_2^2 + \sigma_3^2 + \sigma_4^2}/4$ is used to "com-
5 bine" the four uncertainty ranges. The value $\overline{c_1}$ is consistent with the value of $7.3 \pm 0.06$ MV given in Coleman and Dwyer (2006). And all the estimated values of the $c_2$ and $c_4$ are consistent with each other within a 95 % confidence interval. Combining all the values of $c_2$ and $c_4$ gives :

$$\overline{c_2} = 279 \pm 5.6 \text{ kV/m}$$

$$\overline{c_4} = 288 \pm 4.8 \text{ kV/m}$$

And both value are also consistent with each other, leading to the final value of $\overline{c_{2,4}} = 283.5 \pm 3.69$ kV/m. These values slightly deviates from the value of $276.5 \pm 2.24$ obtained from Coleman and Dwyer (2006) if the values they obtained for the fits of $\lambda$ and $\tau$ are combined. The work of Coleman and Dwyer (2006) used the REAM model too, in a version that should not have significantly changed compared to the one used here. Thus, we think this difference is purely attributed to differences in the methodology that was used to make these estimates from the output data of the code. Concerning the $c_3$ parameter,
combining all the estimates gives $\overline{c_3} = 26.8 \pm 0.32$ ns MV/m, that is slightly lower than the value of $27.3 \pm 0.1$ ns MV/m of Coleman and Dwyer (2006), but none of the values are consistent within the 95 % confidence interval. For this case, we also think the slight difference can be attributed to differences in methodology. Furthermore, the ratio $\overline{c_1}/\overline{c_3}$ can also be used to determine an average speed of the avalanche $\approx \beta_\parallel c$ along the direction of the electric field (that also corresponds to the $z$ direction), and we can estimate $\beta_\parallel \approx 0.90$, that is very close to what was found in previous studies.

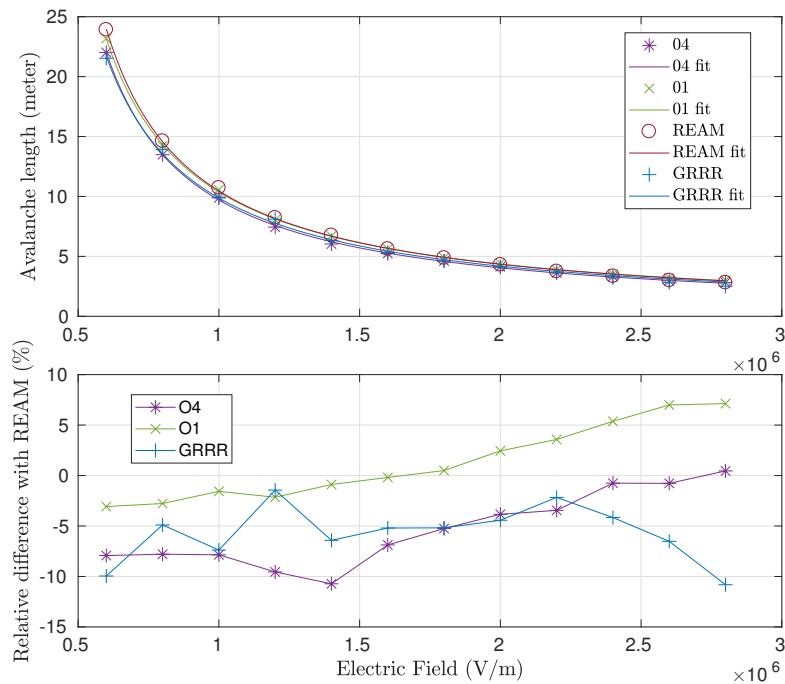

**Figure 3. Top** : Avalanche multiplication length as function of ambient electric field, for each of the codes included in this study. **Bottom** : The relative difference of all other models with respect to REAM. Table 1 indicates the values of the fit parameters.

## 4.2 Evolution to self-similar state

The photon and electron energy spectra of a relativistic runaway electron avalanche (RREA) is known to converge in time to a self-similar solution, where its shape is not evolving anymore, even if the number of particles continues growing exponentially. It may also be referred as the "self-sustained state", or the "steady state" in the literature. At least 5 avalanche lengths (or avalanche times) are required to be able to assert that this state is reached. We propose to estimate this time by looking at the mean electron energy evolution as a function of time. Notice that, as already mentioned in the beginning of Section 4, this mean energy recorded in time is different from the one recorded in distance, used in the next section. We arbitrarily choose to evaluate this mean by averaging all the energies of each individually recorded electron from 10 keV and above. This choice of a 10 keV energy threshold (instead of a higher value, like 511 keV or 1 MeV) does not affect significantly the final estimate of this time to self-similar state. We started with a mono-energetic beam of 100 keV electrons, which is considered low enough compared to the self-similar state mean energy of 6 to 9 MeV. To define the time to self-similar state ($T_s$), we fitted the time evolution of the mean electron energy $\bar{\epsilon}$ with the model

$$\bar{\epsilon}(t) = b_1 - b_2 \times \exp(-t/b_3), \tag{19}$$

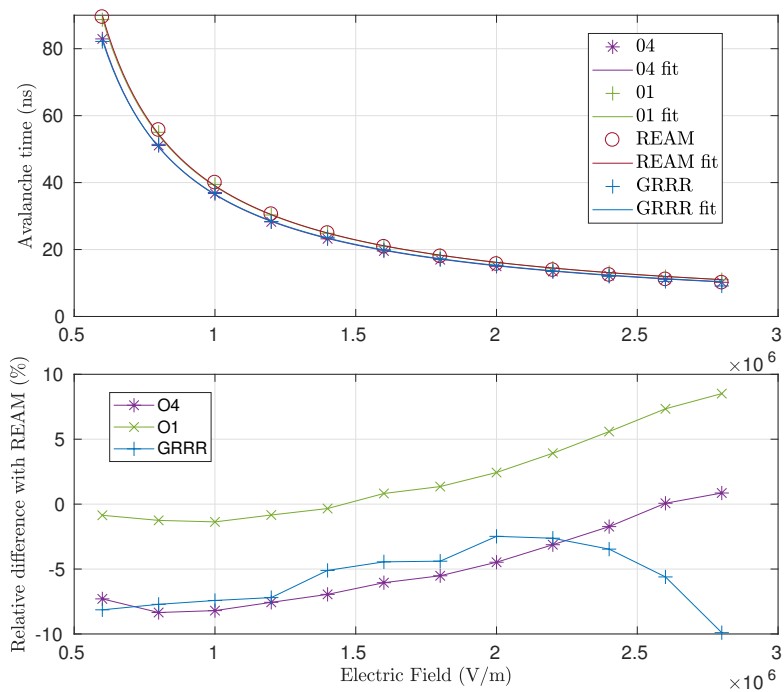

**Figure 4. Top** : Avalanche multiplication time as function of ambient electric field, for each of the codes included in this study. **Bottom** : The relative difference of all other models with respect to REAM. Table 1 indicates the values of the fit parameters.

where $b_1$ and $b_2$ have dimension of energy, $b_3$ dimension of time, and we define $T_s = 5\, b_3$, that is five e-folding times, i.e. converged to 99.3%. The evolution of electron spectra to self-similar state are illustrated for the Geant4 O4 model in the supplementary material (section 12.4). The values of $T_s$ we estimated for the different models are presented in figure 5, together with relative differences of the models with respect to REAM. The relatively high uncertainty (within 95 % confidence intervals) that can be seen on the estimate of $T_s$ is due to a combination of the confidence interval from the exponential fit, from the statistics of the number of seed electrons that could produce a RREA, and from the statistics of the particle counts. For most case, 200 initial seed were used, but for REAM, only 16 seeds were simulated for $E \geq 2.2$ MV/m, and for GRRR, only 20 seeds were simulated above $E \geq 2.0$ MV/m, because of computation time limitations.

In figure 5, Geant O1, O4, GRRR and REAM show consistent times to reach the self-similar state, for all the E-fields. Notice that for them, $T(= T_s/5)$ is close to the avalanche time value $\tau$ given in the top panel of figure 4. For the low electric field of $0.60$ MV/m, it seems to take about 5 times more to reach self-similar state. For this field, there were only three electrons seeds that could produce a RREA, giving a large uncertainty on the estimate of $T_s$, making it impossible to conclude on an inconsistency. From $0.60$ MV/m to $1.8$ MV/m, where all data from codes have good statistics, the times to self-similar state are consistent. From $2.0$ MV/m to $2.4$ MV/m, the two Geant4 models and REAM are consistent, but GRRR present lower times

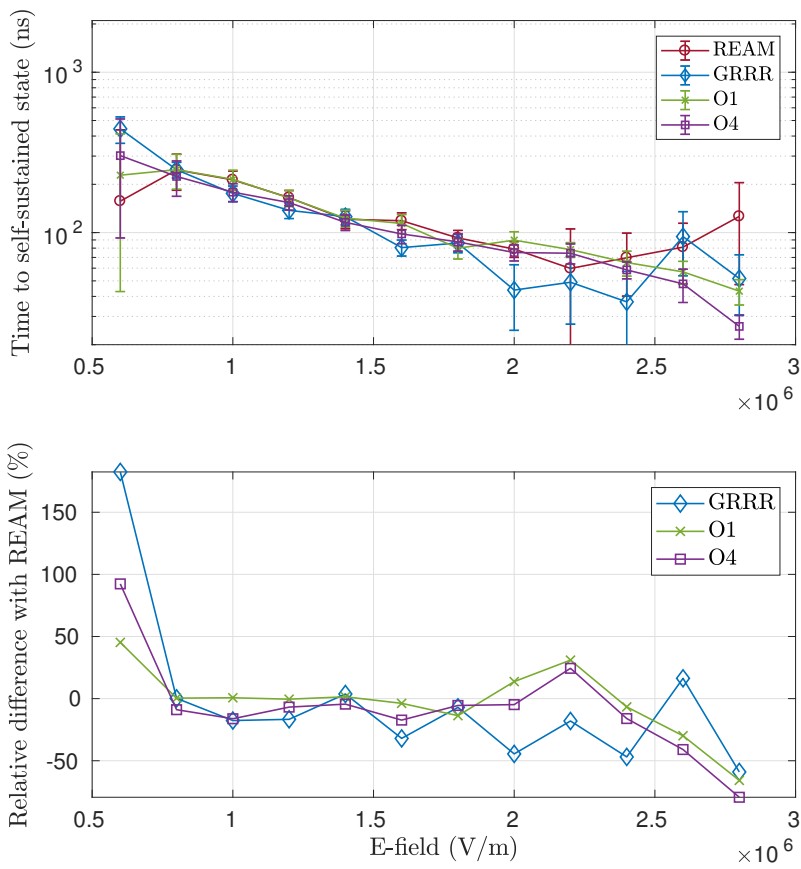

**Figure 5. Top** : time to self-similar state as function of ambient electric field, for each of the codes included in this study. **Bottom** : relative difference with respect to REAM.

by about -20% to -50 %, but it is impossible to conclude an inconsistency, given the large confidence intervals. For E-field magnitudes of 2.6 MV/m to 2.8 MV/m, O1 and O4 present times to self-similar state lower than REAM by about 50 %, that is significant given the uncertainty intervals, whereas GRRR and REAM are consistent. We could not find a clear explanation for it.

5  ### 4.3   RREA spectra

The supplementary material (section 6) presents all the comparison spectra we obtained for photons, electrons and positrons, for the electric field between 0.60 MV/m and 3.0 MV/m. In this section, we discuss the most important differences we could find between the four models.

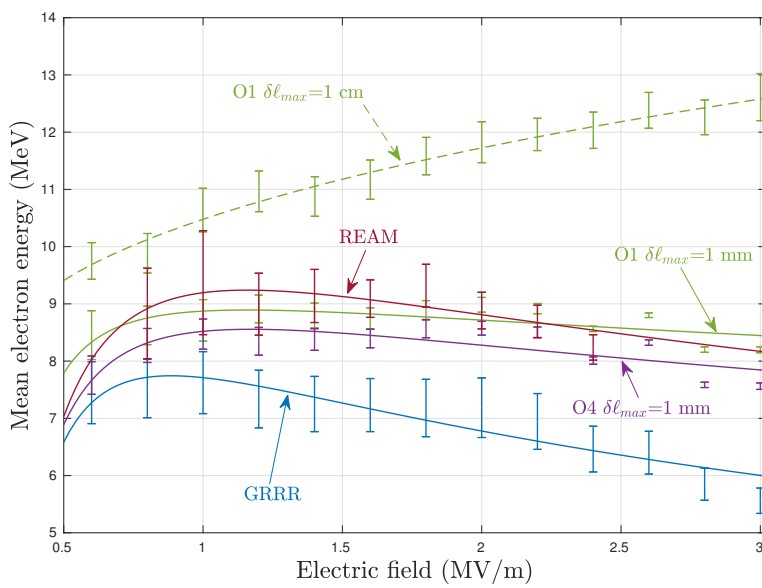

**Figure 6.** Mean electron energies at self-similar state (for distance record), for different electric field magnitudes. The data points are fitted with the model presented in section 4.3.1, equation 20. The values of the fitted parameters are presented in Table 2. To highlight the importance of including step limitations, Geant4 O1 values are presented for two different max step ($\delta\ell_{\max}$) settings: one that is not acceptable (1 cm) and one that is acceptable (1 mm). The parameter $\alpha_R$ is set to its default value of 0.8 for O1 and 0.2 for O4.

### 4.3.1 Electrons

After the RREA electron spectra has reached self-similar state (that requires at least 5 avalanche lengths or times), we recorded the energy spectrum in a plane at a given distance (that is different for each electric field). Then we fitted it with an exponential spectrum model $\propto \exp(-\epsilon/\bar{\epsilon})$ (see also equation 8). Note that for an exponential distribution, the mean of the energy distribution is an estimator of its parameter $\bar{\epsilon}$, justifying the bar notation. We chose to evaluate the mean energy $\bar{\epsilon}$ for record at distances because, contrary to time records, it produces spectra that can be perfectly fit with an exponential distribution over the whole energy range (0 to 100 MeV). Therefore, in this case only the mean RREA electron energy is uniquely defined, and does not depend on an arbitrarily chosen energy threshold, or fitting method. The mean energy $\bar{\epsilon}$ of the exponential spectrum is calculated for the several codes as a function of electric field $E$, as presented in figure 6. For Geant4 O1 the whole simulations and analysis were done twice, for maximum allowed step length settings of $\delta\ell_{\max} = 1$ cm and $\delta\ell_{\max} = 1$ mm, to show that the first case generates totally incorrect spectra, that is consistent with having incorrect RREA probabilities (presented in section 3). In addition, values of the mean energy $\bar{\epsilon}$ for O1 with $\alpha_R = 1.0 \times 10^{-3}$ and $\delta\ell_{\max} = 1$ cm are presented in the supplementary material, section 7.

**Table 2.** Mean energy variation with electric field. For evaluated codes we fitted by equation 20, with $F = 0.28$ MV/m. Figure 6 shows the corresponding curves.

| Parameter / Code | $a_1$ $[10^6\,\text{s}^{-1}]$ | $a_2$ | $a_3$ $[10^6\,\text{s}^{-1}]$ |
|---|---|---|---|
| Geant4 O1 ($\delta l_{max} = 1$ mm) | $6.17 \pm 2.15$ | $1.14 \pm 7.3 \times 10^{-2}$ | $-4.31 \pm 2.0$ |
| Geant4 O4 ($\delta l_{max} = 1$ mm) | $5.17 \pm 1.8$ | $1.23 \pm 8.2 \times 10^{-2}$ | $-1.93 \pm 1.5$ |
| Geant4 O1 ($\delta l_{max} = 1$ cm) | $10.8 \pm 3.4$ | $0.782 \pm 3.9 \times 10^{-2}$ | $-10.7 \pm 3.6$ |
| REAM | $3.98 \pm 2.1$ | $1.31 \pm 0.20$ | $-8.41 \times 10^{-2} \pm 2.1$ |
| GRRR | $4.24 \pm 1.6$ | $1.42 \pm 0.11$ | $-0.639 \pm 1.16$ |

The data of figure 6 was fit following the model,

$$\bar{\epsilon}_{\text{fit}}(E) = \lambda(E)(qE - F), \quad \lambda(E) = \beta c \left[ a_1 \left( \frac{qE}{F} \right)^{a_2} + a_3 \right]^{-1}, \tag{20}$$

motivated by the facts that $\epsilon_2^{\min}$ is roughly a power-law of $E$ (see figure 2) and $\lambda$ is a power-law of $\epsilon_2^{\min}$ (see equation 3). It has three adjustable parameters $a_1$, $a_2$ and $a_3$. We set $F = 0.28$ MeV/m, that is approximately the RREA threshold. The speed $\beta$ is set constant, equal to 0.90, because the RREA velocity does not change of more than 5 % over the range of electric fields we tested. This model is in general agreement with the calculations of Celestin et al. (2012), where $\lambda(E)$ presents an approximately linear relation with the electric field. Table 2 gives the parameters' best fits (with confidence intervals) for the different models, and figure 6 shows the corresponding curves.

In figure 6, it is clear that the Geant4 O1 model with $\delta\ell_{\max} = 1$ cm presents a significantly higher $\bar{\epsilon}(E)$ than the other codes, with values ranging from 9.5 MeV to 12.5 MeV. From the previous RREA probability simulations (see section 3), we know that this $\delta\ell_{\max}$ parameter is not low enough, and so the results of this model can be disqualified. However, when $\delta\ell_{\max}$ is reduced to 1 mm, the results of both Geant4 model are close. There seems to be a consensus between Geant4 (O1 and O4) and REAM, that gives a mean energy that is between 8 and 9 MeV and can vary up to 10 % depending on the electric field. For all electric field magnitudes, GRRR shows a smaller average energy, from about 10 % less at 1 MV/m to about 20 % less at 2.8 MV/m. The reason is certainly because GRRR only includes radiative energy looses as a continuous friction. This is actually a similar difference to what has been observed and discussed in Rutjes et al. (2016) concerning the high energy electron beams, and one can read the discussion therein for more details.

Figure 7 compares the electron spectra recorded at $z = 128$ meters (the electric field has a non-null component only in the $z$ direction, so that electrons are accelerated towards positive $z$), for an electric field magnitude $E = 0.80$ MV/m, for a RREA generated from 200 initial ("seed") electrons with $\epsilon = 100$ keV. This record distance was chosen because it corresponds to about 8.5 avalanche lengths, giving a maximum multiplication factor of about 5000, for which there is not doubt the RREA

is fully developed and has reached self-similar state. This electric field of $E = 0.80$ MV/m was chosen because it is were we could observe the most interesting differences between the models, and it also happens to be the lowest for which we could build spectra with enough statistics on all the models to be able to present a precise comparison. The choice of 200 initial electrons is purely due to computational limitations.

In Figure 7, the error bars on the bottom panel represent the uncertainty due to the Poisson statistics inherent when counting particles. The four models are consistent within 10 % between 20 keV and 7 MeV. Below 20 keV, we think the discrepancy is not physical, and can be attributed to the recording methods set up for the different codes, that are not perfect and have a more or less important uncertainty range (that is not included in the display errors bars, only based on Poisson statistics). Above 7 MeV, O1 remains consistent with REAM overall, but O4 and O1 deviate significantly : up to 50 % for O4 and up to 90 % for GRRR.

For the last bin between 58 and 74 MeV, O4 and GRRR are inconsistent, that is explained by the fact that GRRR does not include straggling for Bremsstrahlung (i.e. either explicit bremsstrahlung collision or some stochastic fluctuations mimicking straggling). The deviations for the high energy part (>7 MeV) in the electron spectrum are significant for this particular field ($E = 0.80$ MV/m), however this is not true for all electric fields, where the codes are overall roughly consistent, as seen in the Supplementary Material (section 6). In principle O4 should be more precise than O1 (Allison et al., 2006), as it includes more

advanced models, yet we cannot argue that O4 is more accurate than REAM. One way of deciding which model is the most accurate might be to compare these results with experimental measurements. but in the context of TGFs and Gamma-ray glows it is complicated to get a proper measurement of electron spectra produced by RREA. However, photons have much longer attenuation lengths than electron and can be more easily detected, e.g. from mountains, planes, balloons or satellites. In the next section we present and discuss the corresponding photon spectra.

**4.3.2   Photons**

In figure 8, the photon spectra recorded at $z = 128$ m (the electric field has a non-null component only in the $z$ direction) for a magnitude $E = 0.80$ MV/m are given for Geant4 O1/O4 and REAM, together with the relative difference with respect to REAM. The reasons why these $z$ and $E$ values were chosen is given in the previous section.

     The error bars in the relative differences represent the uncertainty due to the inherent Poisson statistics when evaluating

particle counts. The Geant4 O1 and O4 models are consistent for the full energy range, except a small discrepancy below 20 keV, that can be attributed to different physical models, O4 being more accurate in principle. In this case, it cannot be attributed to recording methods, since they are exactly the same for both Geant4 models. At 10 keV the two Geant4 spectra are about 80 % larger than REAM. With increasing energy, the discrepancy reduces and reaches 0 % at 100 keV. Above 100 keV, the three models show consistent spectra. There may be some discrepancy above 30 MeV, but it is hard to conclude since the uncertainty

interval is relatively large.

     As just presented, the main noticeable discrepancy between O1/O4 and REAM is present below 100 keV. As far as we know, there is no reason to argue that Geant4 gives a better result than REAM in this range, or vice-versa. One way to find out which model is the most accurate could be to compare these results with real measurements. Are such measurement possible to obtain? Any photon that an instrument could detect has to travel in a significant amount of air before reaching detectors. The

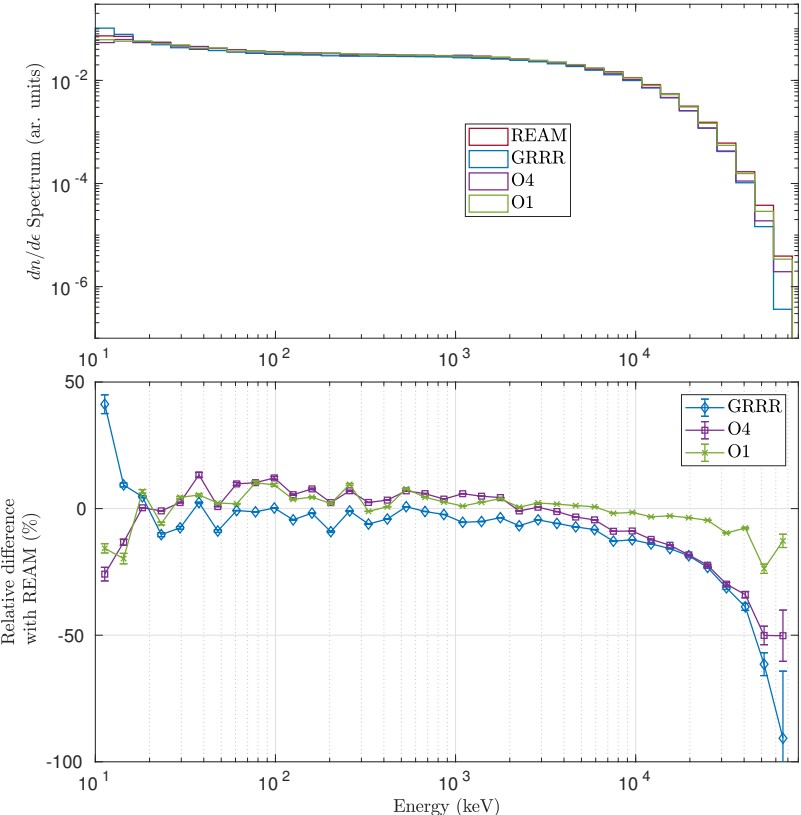

**Figure 7. Top** : Electron (kinetic) energy spectra of Geant4 (O4 and O1), REAM and GRRR, for $E = 0.80$ MV/m, recorded at $z = 128$ m. The RREA is generated from 200 seed electrons of $\epsilon = 100$ keV. **Bottom** : relative difference between REAM and the three other models. The error bars are calculated from the Poisson statistics.

average path traveled in the atmosphere by a 100 keV photon in 12 km altitude air is $1540 \pm 806$ meters. It decreases for lower energies and is $671 \pm 484$ meters at 50 keV, and $63.0 \pm 61.5$ meters at 20 keV. Note that these lengths have been evaluated from precise Geant4 simulations, and are smaller than the attenuation lengths at the same energies, because photons gradually loose energy due to stochastic collisions. These average traveled paths are too small for the photons to have a reasonable chance
5 to escape the atmosphere and to be detected by a satellite. But we cannot exclude that they may reach an airborne detector located inside or close to a thunderstorm. As a side note, we want to indicate that the vast majority (if not all) of the photons observed from space with energies below a few hundred of kilo-electronvolts (e.g., by the Fermi space telescope, see Mailyan et al. (2016)) had very likely more than 1 MeV when they were emitted. They lost some part of their energy by collisions (with air molecules in the atmosphere or/and with some part of the satellite) before being detected by the satellite. For information,
10 a figure presenting the probability of a photon to escape the atmosphere as function of its primary energy for a typical TGF is presented in the supplementary material, section 14.

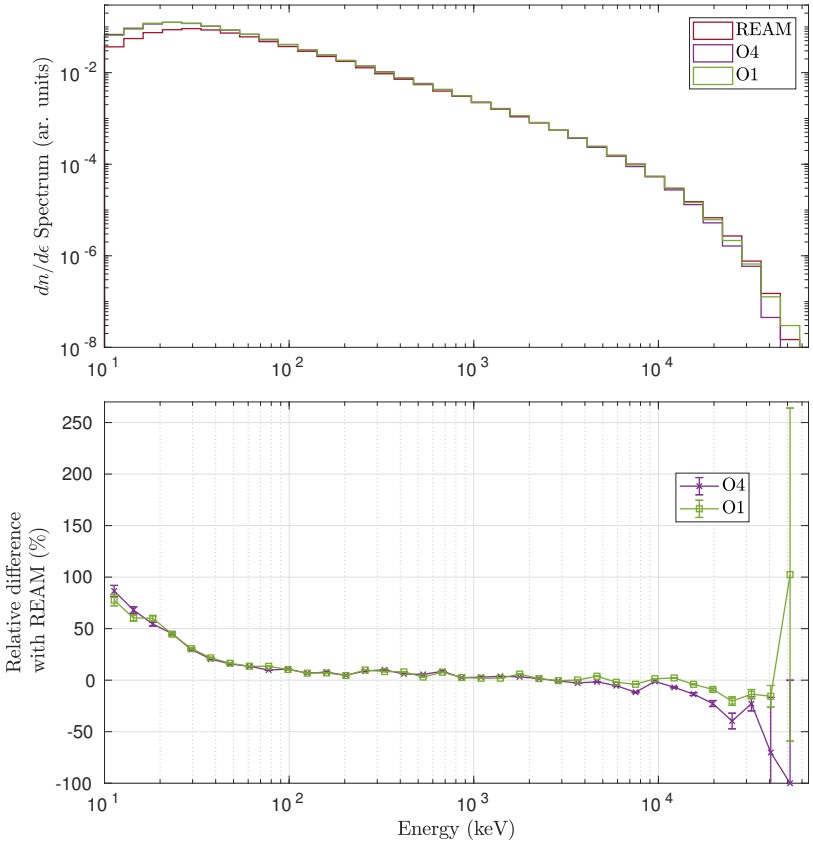

**Figure 8. Top** : Photon energy spectra of Geant4 (O4 and O1) and REAM for $E = 0.80$ MV/m, recorded at $z = 128$ $m$. **Bottom** : relative difference between Geant4 (O1 and O4) and REAM. The error bars are calculated from the Poisson statistics.

### 4.4 Other differences

In addition to what is presented so far in this article, the following points should also be mentioned when comparing the results of the codes. The corresponding plots are available in the Supplementary Material.

– The mean parallel (to the E-field direction) velocity $\beta_\parallel$ of the avalanche is shown in section 4.2 of the Supplementary
5      Material (labeled "mean Z velocity"). We observe that GRRR is giving $\beta_\parallel$ faster than all the other codes, and O4 is systematically slower than REAM and O1, though the differences are less than 2 %. The variation of $\beta_\parallel$ towards the electric field $E$ is small, about 10 % for all codes. For increasing E-fields, electrons are less scattered and more focused in the field direction, hence slightly increasing $\beta_\parallel$.

– The electron to (bremsstrahlung) photon ratio $r_{e/p}$ was also calculated and compared for different distance record in the
10     RREA shower, and the corresponding plots are presented in the Supplementary Material, section 3. GRRR is excluded because it does not include photons. For any electric field, the same discrepancy is observed. At the beginning of the

shower (<4 avalanche lengths), $r_{e/p}$ appears to be about 20 % larger for REAM compared to O1 and O4, then the three models are consistent at a given distance, and finally for more than about 4 avalanche lengths, the tendency is inverted and REAM presents a $r_{e/p}$ about 20 % smaller than Geant4. The magnitude of this discrepancy is largely reduced for increasing electric fields. We did not fully understand the reasons of these differences, and it may be due to the bremsstrahlung models used are involved. More investigations are required.

– The positron spectra have relatively low statistics (in the order of few hundreds particles recorded) and are all quite consistent within the relatively large uncertainties.

– In the photon spectra obtained from particle records at fixed times, REAM seems to show significantly less (at least a factor of 10) photon counts than the two Geant4 models for most of the electric fields magnitudes. For some fields, it even shows a lack of high energy photons, with a sharp cut at about 30 MeV. It seems to point out to a problem in the record method, explaining why we chose not to discuss these spectra in the main article. The spectra produced by the Geant4 O1 and O4 models for this case are consistent with one another for all the E-fields.

## 5  Conclusions

We have investigated the results of three Monte Carlo codes able to simulate Relativistic Runaway Electron Avalanches (RREA), including the effects of electric fields up to the classical breakdown field, which is $E_k \approx 3$ MV/m at STP. The Monte-Carlo codes REAM, GRRR and Geant4 (two models: O1 and O4) were compared. The main difference between the Geant4 O4 and O1 models is the inclusion of more precise cross sections for low energy interactions ($< 10$ keV) for O4.

We first proposed a theoretical description of the RREA process, that is based and incremented over previous published works. Our analysis confirmed that the relativistic avalanche is mainly driven by electric fields and the ionisation and scattering processes determining $\epsilon_2^{\min}$, the minimum energy of electrons that can runaway. This is different from some of the previous works that speculated that the low energy threshold ($\varepsilon_c$), when changed from 1 keV to 250 eV, was the most important factor affecting the electron energy spectra (Skeltved et al., 2014; Rutjes et al., 2016).

Then, we estimated the probability to produce a RREA from a given electron energy ($\epsilon$) and a given electric field magnitude ($E$). We found that the stepping methodology is of major importance, and the stepping parameters are not set up satisfactorily in Geant4 by default. We pointed out which settings should be adjusted and provided example codes to the community (see sections 6 and 7). When properly set-up, the two Geant4 models showed a good overall agreement (within $\approx 10$ %) with REAM and GRRR. From the Geant4, GRRR and REAM simulations, we found that the probability for the particles below $\approx 10$ keV to accelerate and participate in the penetrating radiation is actually negligible for the full range of electric field we tested ($E < 3$ MV/m). It results that a reasonable lower boundary of the low energy threshold ($\varepsilon_c$) can be set to $\approx 10$ keV for any electric field below $E_k \approx 3$ MV/m (at STP), making it possible to have relatively fast simulations. For lower electric fields, it is possible to use larger $\varepsilon_c$, following a curve we provided (Figure 2.b).

The advantage of using more sophisticated cross sections able to accurately take into account low energy particles could be probed by comparing directly the O1 and O4 models. They showed minor differences that are mainly visible only for high E-fields ($E > 2$ MV/m), where low energy particles have more chance to runaway.

In a second part, we produced RREA simulations from the four models, and compared the physical characteristics of the produced showers. The two Geant4 models and REAM showed a good agreement on all the parameters we tested. GRRR also showed an overall good agreement with the other codes, except for the electron energy spectra. That is probably because GRRR does not include straggling for the radiative and ionisation energy losses, hence implementing these two processes is of primary importance to produce accurate RREA spectra. By comparing O1 and O4, we also pointed out that including precise modelling of the interactions of particles below $\approx 10$ keV provided only small differences; the most important being a $5\%$ change in the avalanche multiplication times and lengths. We also pointed out a discrepancy from Geant4 (O1 and O4) compared REAM, that is a 10% to 100% relative difference in the low energy part ($< 100$ keV) of the photon energy spectrum for an electric field of $E = 0.80$ MV/m. But we argued that it is unlikely to have an impact on spectra detected from satellites.

## 6   Recommendations

From the experience of this study, we give the following general recommendations concerning RREA simulations :

- Codes should be checked / tested / benchmarked using standard test set-ups. In the supplementary material, we provide a precise description of such tests. In section 7 of this article, we provide links to download the full data-set we obtained for the codes we tested (Geant4 with two set-ups, REAM and GRRR), as well as processing scripts. We also provide the source code of the Geant4 codes.

- Custom-made codes should be make available to other researchers, or at least the results they give for standard tests.

- In order to make it possible to compare results from different studies, the methodology used to derive a given quantity should be rigorously chosen, and presented clearly somewhere.

- Extending the recommendations of Rutjes et al. (2016), we concluded that to get an accurate RREA electron spectra above 10 MeV, radiative loss (bremsstrahlung) should not be implemented with uniform friction only: straggling should be included. Straggling should also be included for ionisation energy loses below the energy threshold.

Concerning the usage of Geant4 for simulating RREA :

- Default settings are not able to simulate RREA accurately. To get accurate RREA results, one of the following tweaks is possible :

  - Changing the $\alpha_R$ ("dR over Range") parameter of the electron/positron ionisation process to $5.0 \times 10^{-3}$ or less. This solution gives the best ratio between accuracy and computation time. Leave the "final range" parameter to one millimeter (default value) or less.

- Setting up a step limitation process (or a maximum acceptable step) to one millimeters or less. This will significantly increase the required computation time.

- Using the single (Coulomb) scattering model instead of multiple scattering (the two previous tweaks relying on the multiple scattering algorithm). This will substantially increase the necessary computation time. This is because multiple scatterings algorithms were invented to make the simulation run faster by permitting to use substantially larger (usually >10 times) step lengths compared to a pure single scattering strategy, while keeping a similar accuracy.

- In section 7, we provide a link to Geant4 example source codes implementing these three methods.

- Compared to using the default Møller/Bhabha scattering models for ionisation, the usage of more accurate cross sections, e.g. taking into account the electrons' molecular binding energies (like done for the Livermore or Penelope models), only leads to minor differences.

## 7 Code and/or data availability

The full simulation output data of the four models is available though the following link:

https://filesender.uninett.no/?s=download&token=738a8663-a457-403a-991e-ae8d3fca3dc3

The scripts used to process this data to make the figures of the supplementary material are available in the following repository:

https://gitlab.com/dsarria/HEAP2_matlab_codes.git

The full GRRR source code is available in the following repository :

https://github.com/aluque/grrr/tree/avalanches

The Geant4 source code for the RREA probability simulations is available in the following repository :

https://gitlab.com/dsarria/av_prob.git

The Geant4 source code for the RREA characterisation simulations is available in the following repository :

https://gitlab.com/dsarria/RREA_characteristics.git

*Author contributions.* DS, CR and GD designed the tests. DS, CR, GD wrote most of the manuscript. DS, GD and CR proceeded to the data analysis and made the figures and tables. DS carried out the Geant4 simulations and provided the data. AL carried out the GRRR simulations and provided the data. JRD and KMAI carried out the REAM simulations and provided the data. NO, KMAI, JRD, UE, ABS and ISF provided important feedback and review on the text.

*Acknowledgements.*

This work was supported by the European Research Council under the European Union's Seventh Framework Program (FP7/2007-2013)/ERC grant agreement n. 320839 and the Research Council of Norway under contracts 208028/F50 and 223252/F50 (CoE). For part of the results of this work, it was necessary to use the Fram computer cluster of the UNINETT Sigma2 AS, under project number NN9526K.

G. D. is supported by the Brazilian agency CAPES. C.R. acknowledges funding by FOM Project No. 12PR3041 that also supported G.D.'s 12 month visit in The Netherlands. I.S.F. thanks CNPqs grant PDE(234529/2014-08), and also FAPDF grant number 0193.000868/2015, 03/2015.

This material is based in part upon work supported by the Air Force Office of Scientific Research under award number FA9550-16-1-0396.

## Appendix A: Geant4 relative performance

Table 3 presents the relative computation times it takes to complete the simulation with an electric field magnitude of 1.2 MV/m, and 100 initial ("seed") electrons with initial energy $\epsilon = 100$ keV, and a stop time (physical) of 233 nanoseconds. The fastest simulation uses Geant4 with the O1 physics list and $\delta\ell_{\max} = 10$ cm and took 4.53 seconds to complete on one thread with the microprocessor we used. The simulations with the O4 physics list with $\delta\ell_{\max} = 1$ mm requires about 400 times more computation time. Setting up $\delta\ell_{\max} = 1$ mm, or lower, is necessary to achieve correct simulation of the RREA process, as argued in section 3. To achieve it for the full range of electric fields we tested (in a reasonable amount of time), it required the use of the Norwegian FRAM computer cluster. The simulations with $\delta\ell_{\max} = 0.1$ mm for all electric fields could not be achieved in a reasonable amount of time, even by using the computer cluster.

On the other hand, if $\delta\ell_{\max}$ is left at its default value (1 kilometer) and $\alpha_R$ parameter is tweaked instead, accurate simulations can be achieved with a value of $\alpha_R = 5.0 \times 10^{-3}$ or lower. It requires almost an order of magnitude less computation time compared to using $\delta\ell_{\max} = 1$ mm.

| | Model | Option 1 (O1) | Option 4 (O4) |
|---|---|---|---|
| $\delta\ell_{\max}$ | | | |
| 10 cm | | **1** | **6.49** |
| 1 cm | | **11.5** | **27.2** |
| 1 mm | | **222** | **393** |
| 0.1 mm | | **2100** | **3740** |
| $\alpha_R$ (default) | | 0.80 | 0.20 |

| | Model | Option 1 (O1) | Option 4 (O4) |
|---|---|---|---|
| $\alpha_R$ | | | |
| 0.80 | | **$\approx$1** | **2.44** |
| 0.20 | | **2.61** | **7.66** |
| 0.050 | | **7.12** | **36.5** |
| 0.0050 | | **21.0** | **126** |
| 0.0010 | | **41.7** | **224** |
| $\delta\ell_{\max}$ (default) | | 1 km | 1 km |

**Table 3.** Computation time needed by different Geant4 configurations for the simulation of the same physical problem, relatively to the Geant4 O1 $\delta\ell_{\max} = 10$ cm case. Two parameters are tested : the maximum allowed step ($\delta\ell_{\max}$) and the "dR over Range" ($\alpha_R$).

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
