# Peer review of "Evaluation of Monte Carlo tools for high energy atmospheric physics II : relativistic runaway electron avalanches"

_Geoscientific Model Development, 2018_

## Referee Comment (RC1) · A Chilingarian (Referee) · 29 Jun 2018

Comments on the paper submitted to Geoscientific Model Development (GMD) "Evaluation of Monte Carlo tools for high energy atmospheric physics II : relativistic runaway electron avalanches" by David Sarria, Casper Rutjes, Gabriel Diniz, et al.

The authors compare simulations made by several M-C codes and come with useful recommendations on simulation procedures (energy cuts, etc..). The analysis is detailed and there are no doubts that all codes in own limits produce rather coherent results. The code verification and comparisons of different code options, as well as different codes, are necessary first step of simulation experiments and constructing of

models to be compared with experiment.

However, we have to understand, that M-C simulations for such a complicated domain as High-energy Physics in Atmosphere (HEPA) is not a precise tool! We don't know the distribution of the electric charges in the cloud and, therefore, strength and elongation of the emerging electric fields. Therefore, very time-consuming and detailed verification of different M-C programs, for opinion of this referee is not too important on the present stage of HEPA progress. As I mention in my review to the first paper of this series in 2016, the validation of the available experimental observations is vital for the progress of HEPA. There are published numerous gamma ray energy spectra observed on the mountain altitudes and few electron energy spectra; why not to try to compare simulations with observations? Continuous simulations with different codes and arbitrary parameters (sometimes nonrealistic, see my comment below to 4.3.1) can make illusion of intense scientific research; however only comparisons with observations and physical inference on the observed phenomena really values. Sure, authors will argue that model validation is out of scope of their paper. And they will be right. However, I can ask, when they will use their verified models for coming with comprehensive model of HEPA? When they will develop models with realistic parameters and compare it with data (energy spectra measured on Earth's surface and in the space)?

To be not too didactic, I'll cite our old paper, where we try for the first time to compare simulated energy spectra with measured ones and establish a TGE model (see Figs 8-12 of Chilingarian, Mailyan and Vanyan, 2012).

"With newly estimated thundercloud height, we re-estimate several phenomenological parameters of the RREA process as the following: the most probable height of thundercloud (and electrical field therein) is $\sim$50 m. The number of electrons with energies above 1 MeV at the exit from the cloud is 1.97âĹŮ107 electrons/m2/min; if we assume that the radiation region in the thundercloud has a radius of 1 km the total number of electrons crossing this region in a minute is $\sim$6âĹŮ1013.

Sure, we use not optimized M-C; and, maybe we make some mistakes in our inference. We discuss possible sources of the systematic errors:

"We do not measure the electric field within the thundercloud; near surface electric field is not a good proxy of the intracloud fields accelerating electrons downward. We also do not measure vertical extension of the field and only estimate the height of the cloud. Therefore, simulations of the RREA process in the atmosphere with chosen parameters, although are in an agreement with the available measurements of electric fields in the thunderclouds, cannot be used for direct comparisons with TGE measurements. However, these simulations give us understanding of the RREA scale and MOS processes and expected behavior of the energy spectra."

However, it was the first time we present gamma ray and electron energy spectra along with simulations and achieve overall agreement. Now we develop a new method of cloud height estimation and can approach observations with more realistic simulations with more reliable better parameters. What I want to demonstrate is that simulation should be paired with experimentation; and each should profit from other.

After our recently observation of Long Lasting Low energy TGEs (LLL TGEs) – a hours extending flux of gamma rays of 0.3 – 3 MeV energies, we started a cycle of simulations to get answer if remote Extensive Cloud Showers (ECSs) can contribute to this flux, or we should consider stochastic electron acceleration by a "sea" of randomly distributed charges in the thundercloud. Thus, simulations are pairing with observations and with hypothesis testing.

The emerging field of High Energy Atmospheric Physics studies events producing high energy particles and associated with thunderstorms, such as terrestrial gamma-ray flashes and gamma-ray glows. Without mentioning Thunderstorm Ground Enhancements (TGEs) this statement is biased.

The difference in duration between TGF and gamma-ray glows can be explained by two possible different scenario to create runaway electrons... Largest TGE detection prove that long duration can be explained by the continuous acceleration and multi-plication of seed electrons entering strong prolonged electric field. Such a condition can sustain for minutes and, so called, extensive cloud showers (ECSs) will produce fluxes of electrons, gamma rays and neutrons on the earth's surface, i.e. TGEs (see Chilingarian et al., 2017). 1.3 5 The physics behind TGF, TGF afterglows and gamma-ray glows are studied with the help of computer simulations, which necessarily involves model reduction and assumptions. Hopefully physics is experimental science and most of results are obtained by experimentation, not simulation. 4.3.1 5 Figure 7 compares the electron spectra recorded at 128 meters, for an electric field E = 0.80 MV/m, for a RREA generated from 200 seed electrons with $\varepsilon$ = 100 keV. Do you especially choose the field never measured in the atmosphere (0.8 MV/m)? Or it is not in atmosphere? Why 128 m? Where do you inject 200 electrons?

15 One way of deciding which model is the most accurate might be to compare these results with experimental measurements. but in the context of TGF and Gamma-ray glows it is complicated to get a proper measurement of electron spectra produced by RREA. Finally, yes, only way to decide which model is true is the comparison with experiment that is missing in this paper.

References

Chilingarian, B., Mailyan, Vanyan, L., 2012. Recovering of the energy spectra of electrons and gamma rays coming from the thunderclouds. Atmos. Res. 114–115, 1–16. Chilingarian A., Hovsepyan G., Mailyan B., 2017, In situ measurements of the Runaway Breakdown (RB) on Aragats mountain, Nuclear Inst. and Methods in Physics Research, A 874,19–27.

---

## Short Comment (SC1) · 1 Jul 2018

Dear colleague

Link on code with geant4 and other don't work. Only HEAP2$_{matlab_codes}$ link is correct.

The scripts used to process this data to make the figure of the supplementary material are available in the following reposi- tory: https://gitlab.com/dsarria/HEAP2$_{matlab_codes}$.git The full GRRR source code is available in the following repository : $https://github.com/aluque/grrr/tree/avalanches$ The Geant4 source code for the RREA probability simulations is available https://gitlab.com/dsarria/av$_{p}$rob.git The Geant4 source code for the RREA characterisation simulations is available in the f

$https://gitlab.com/dsarria/RREA_characteristics.git$

---

## Short Comment (SC2) · 1 Jul 2018

Dear colleague,

Thanks for you comment. The link problems should be fixed now.

For the two Geant4 source codes :

https://gitlab.com/dsarria/av_prob.git

https://gitlab.com/dsarria/RREA_characteristics.git

The access was not set to public yet. If you still have problems, please send me an

e-mail and I will share the codes.

The link to the GRRR GitHub's deposit (A. Luque's code) should have always been working.

https://github.com/aluque/grrr/tree/avalanches

I tested it from several computers, with several web browsers.

Maybe access is restricted for Russia. In this case, contact Alejandro Luque: aluque@iaa.es .

Best regards, -David Sarria
* * *

---

## Referee Comment (RC2) · Anonymous Referee #2 · 26 Jul 2018

General comments:

The authors compare three different codes (Geant 4 with two different input parameters, GRRR and REAM) with respect to the formation of relativistic runaway electron avalanches (RREA). They compare fundamental properties such as the avalanche length, the mean electron energy or the photon energy distribution. They find that these three codes show a good agreement with each other. Where they do not, the authors try to elaborate which parameters might be responsible for the deviation. Finally, the authors give a recommendations regaring RREA simulations. In my opinion, this paper is valuable for the community. However, before publication, I have a few

suggestions on the presentation; I also think that some parts need further clarification.

Specific comments:

Abstract:

The authors mention the "effects of electric fields" (l. 5). I assume, this study is only about ambient fields and does not include self-consistent electric fields by solving the Poisson equation. Also, in line 10, it would already good to mention what kind of "stepping methodology" (line 10) is meant, i.e. of particles. The authors mention that they only tested electric fields until 3 MV/m; however, the electric field for thermal runaway, i.e. for all electrons to run away irrespective of their initial energy, is approx. 26 MV/m, why not consider fields between 3 MV/m and 26 MV/m?

1. Introduction:

1.1:

line 4: If it is about observations of high-energy phenomena, I would suggest to cite Fishman et al., 1994 in addition to Williams, 2010.

1.2:

In line 14, the authors talk about the energy regime of HEAP. However, they do not define this energy regime. Please be more precise in defining the energy range.

Futhermore, the authors write that some inidividual electrons do not survive. Especially, for high-energy electron beam, it would be good to name the reason for this. Also, please specify what values you consider "much larger than the ionisation threshold" (page 3, lines 23-24). On page 4, lines 14-15, the authors write "The minimum energy $\epsilon_2^{min}$ that can runaway is given by the requirement $F(\epsilon_2^{min}) > E$ [...]." But should the friction force not be smaller for runaway. Please clarify this. On page 5, the authors discuss the angular dependency (between the electron motion and electric field direction) on the run-away process. This has already been discussd very extensively by [O.

Chanrion et al., 2016. Influence of the angular scattering of electrons on the runaway threshold in air. Plasma Phys. Contr. Fus., vol. 58, 044001]. Please cite this article.

2. Model descriptions:

2.1:

The authors say that different sets of electro-magnetic cross sections are used. However, the authors do not state (neither in the main text nor in the supplementary material) which processes are actually taken into account. This is clearly missing, but crucial since simulation results strongly depend on the chosen processes and cross sections. The authors state where the cross sections come from and they make some comparison plots in section 8 of the supplementary material, but only for a few processes. Say, in the future, other researchers want to compare their results. Then, the knowledge of the used cross section data is crucial to interpret results. I would thus suggest to elaborate more on the processes and cross sections.

2.2:

On page 8/line 29, the authors say that GRRR uses the "energy at that instant" to calculate the collision rate $\nu_k$. However, it is not clear which energy: the energy of each individual particle, the maximum energy of all particles or the mean energy of all particles. Please clarify.

2.4.1:

For space-oriented codes, "a single particle is simulated over its entire life-time" (page 9/line 21). However, what is the reason to lose an electron (especially in the regime above several keV). The only reason to lose an electron would be attachment to air molecules. However, for this process to occur the electron normally needs to lose more energy than down to 10 keV. Please be more precise here.

2.4.2:

Why would "acceptable values of $\delta\ell_{max}$ depend on the electric field" (page 10/line 28)? It is very clear that it should be smaller than the electron's mean free path. But the mean free path depends on the electron energy rather than on the electric field. Please clarify the dependence on the electric field.

3. Probability of generating RREA:

In Figure 1, the authors present the probability that a single electron with an initial energy of 75 keV in an ambient field of 0.8 MV/m creates an RREA. As a criterion, they use that at least 20 electrons with an energy of 1 MeV are created. Please specify which value for $\delta\ell_{max}$ is used in panel a) and which value for $\alpha_R$ is used in panel b). However, I am confused, though. The friction force based on [A.V. Gurevich, 1961. On the theory of runaway electrons. Sov. Phys. JETP-USSR, vol. 12, pp. 904–912] is supposed to be 0.65 MeV/m for a 75 keV electron, thus the ambient field is definitely sufficient to accelerate the electron into the run-away regime. Of course, this does not mean that 20 electrons with energies are above 1 MeV, but 12% seems low. But this might depend on the simulation time. How long has the shower been simulated? It would also be good to see the RREA percentage for different criteria (20 electrons above 1 MeV, 10 electrons above 1 MeV, 5 electrons and finally 1 electron above 1 MeV which should give almost 100%). In Figure 2, the authors show the avalanche probabilities (10%, 50% and 90%) as a function of initial electron energy and ambient electric field. What about the right top (high energy, high field) and bottom left (low energy, low field) part? What are the probabilities there? There is so much space in this figure. Why not add some curves or values for these two regimes.

4. Characterisation of RREA showers:

4.2:

When discussing the evolution of the self-similar state , the authors say that they used a different number of seed electrons for Geant4, REAM and GRRR (page 16, lines 12–13). For consistency, I suggest to add one more case where the same number of

seed electrons is used.

Figure 5 shows the time to reach the self-similar state. Comparing all the different models, it seems that the time to reach that is consistent within one order of magnitude. I propose to add this to line 14 on page 16.

4.3.1:

In Figure 6, the authors present the mean energy (of the self-similar state) as a function of the ambient field. It might good to remind the reader in the figure caption which value for $\alpha_R$ was used here for O1 andd O4. I am wondering how the plots for O1 and O4 would look for $\alpha_R$. It would be good to plot one case for a different $\alpha_R$ to show the dependence of the mean electron energy as a function of $\alpha_R$.

In Table 2, the authors present the fit parameters $a_1$, $a_2$ and $a_3$ of Eq. (20) for different models. Please add the error bars in order to judge the quality of these fits.

In Figure 7, the authors present the electron spectra at 128 m. But does 128 m refer to the z-coordinate or to the travelled distance $r = (x^2 + y^2 + z^2)^{1/2}$. Please clarify this in the figure caption and in line 5 (page 19), line 2 (page 20) and in the caption of Figure 8.

4.3.2:

In section 4.3.2, the authors state that a comparison with photon measurements is difficult because of the attenuation of photons in air. Whereas I agree in general, there are some issues I would like to address. The authors say that a 100 keV photon at 12 km altitude travels 1540 m in average, a 50 keV photon 671 m and a 20 keV photon 63 m. Where do these values come from? Is this a result of their simulations (if so, how did you obtain these); if not, please cite your source. I made a brief comparison with NIST data (http://physics.nist.gov/PhysRefData/Xcom/Text/XCOM.html) and obtained attenuation lengths of approx. 2000 m for 100 keV, 1600 m for 50 keV and 500 m for 20 keV. This needs to be clarified. Additionally, the authors say that the "photons

have no chance to escape the atmosphere and to be detected by a satellite" (pages 20/21). I would like to remind the authors that the "average path travelled" (page 20/line 14) or the attenuation length is only an average. Hence, there can always be photons which espace the atmosphere and may be detected at satellite altitudes even though the probability is low. Actually, Fermi has measured photons with energies between 10 and 500 keV (see https://gammaray.nsstc.nasa.gov/gbm/science/terr_grf.html). Please be more precise here.

4.4:

In the supplementary material, section 9.2., I cannot find any plot showing the parallel velocity $\beta_{\parallel}$ (only the mean Z speed; or is the mean Z speed meant to be $\beta_{\parallel}$). Maybe, it is there, but at least not apparent. Could the authors please point me to the correct plot?

Technical corrections:

Abstracts:

line 2: "particles and associated" → "particles associated"

1. Introduction:

1.1: page2/line 12 and overall the manuscript: It is rather common to abbreviate "Terrestrial gamma-ray flashes" with TGFs than with TGF.

page 2/lines 29/30: Köhn and Ebert, 2015. also discuss electron acceleration in the vicinity of lightning leaders instead of from streamer tips.

page 2/line 30: Kohn → Köhn

page 2/line 34: A fullstop is missing

page 3/line 6: lighting → lightning

1.2:

[Figure]

page 3/line 22: keeps → keep

page 4/line 13: 1-dimension → one dimension

page 5/line 5: equation → equations

1.3:

page 6/line 19: consists in → consists of

page 6/line 28: that energy the spectrum → that the energy spectrum

1.4:

page 7/line 3: Kohn et al. → Köhn et al.

2. Model description:

2.1:

page 8/line 15: this parameters was thought to be responsible major change → this parameter was thought to be responsible for a major change

2.4.1:

page 9/line 26: loose → lose

2.4.2:

page 10/line 13: in previous section → in the previous section

3. Probability of generating RREA:

page 11/line 29: as function → as a function of

page 12/line 1: for O1 → for O1 and O4

4. Characterisations of RREA showers:

4.1:

page 13/line 4: Figure → Figures

page 13/line 14: s. V/m → s· V/m

page 14/line 14: value → values

4.2:

page 14/line 24: spectra of relativistic → spectra of a relativistic

page 14/line 28: looking to the mean electron energy evolution as function of time → looking at the mean electron energy evolution as a function of time

page 14/line 30: all the energy of each individually recorded electrons → all the energies of each individually recorded electron

page 16/line 2: are → is

page 16/line 7: electrons → electron

page 16/line 20: conclude to an → conclude an

4.3 (page 17):

line 2: for photon → for photons

4.3.2:

page 20/line 2: O4/O4 → O1/O4

page 20/line 9: large large → large

6. Recommendations (page 23):

line 15: looses → loses

line 19: Maybe it s a good idea to define $\alpha_R$ as "dR over Range" earlier in the manuscript.
[Figure]

lines 24/25: The authors say that single Coulomb scattering would "increase the necessary computation time". Should using a single scattering (instead of multiple scattering) not decrease the computation time?

line 26: we provide link → we provide a link

7. Code and/or data availability (page 24):

line 4: figure → figures

Appendix A: Geant4 relative performance:

page 24/line 24: Table A1 → Table 3 (Or change table 3 to table A1)

page 24/line 24: Electric field → electric field

Caption of Figure 3 (page 25): dRoverRange → dR over Range

---

## Author Comment (AC1) · 13 Sep 2018

See attached "gmd-2018-119-supplement.zip" file containing the following .pdf files :

- response_referee1.pdf: the response to the referee 1

- sarria_GMD_highlight_diff.pdf: the revised manuscript where the differences with the first draft manuscript are highlighted (taking into account comments of both referees)

- sarria_GMD_revised_article.pdf: the revised manuscript

---

## Author Comment (AC2) · 13 Sep 2018

See attached "gmd-2018-119-supplement.zip" file containing the following .pdf files :

- response_referee2.pdf: the response to the referee 2

- sarria_GMD_highlight_diff.pdf: the revised manuscript where the differences with the first draft manuscript are highlighted (taking into account comments of both referees)

- sarria_GMD_revised_article.pdf: the revised manuscript

---

## Author Response (AR2)

**Author's Response**

**Article:** Evaluation of Monte Carlo tools for high energy atmospheric physics II : relativistic runaway electron avalanches
**by D. Sarria, C. Rutjes, et al.**

**Journal:** Geoscientific model development (GMD)
**Manuscript reference:** gmd-2018-119

**Contents**

**1 Response to the review of Referee # 1 (with relevant changes listed with italic font)**

**Response to referee #1**

We thank the Referee for his careful reading, and the valuable comments. We considered each of the comments and questions and provided an adequate answer. The comments of Referee #1 are written in bold and the answers in plain text. Sentences indicating a modification to the manuscript are written using italic font.

The latex diff of the revised manuscript, accounting for comments of referees #1 and #2 is provided as an attached file.

**General comment**

**The authors compare simulations made by several M-C codes and come with useful recommendations on simulation procedures (energy cuts, etc..). The analysis is detailed and there are no doubts that all codes in own limits produce rather coherent results. The code verification and comparisons of different code options, as well as different codes, are necessary first step of simulation experiments and constructing of models to be compared with experiment.**

We would like to clarify that we do not intend to validate/verify any code in this article. The general purpose code Geant4 has already undergone multiple verification and validation studies in several physical contexts, but nothing guarantees it can be straightforwardly applied to HEAP (also referred as HEPA by the referee) phenomena, in particular concerning the capacity of simulating Relativistic Runaway Electron Avalanches (RREA) that we are extensively testing here. This issue was first raised by (*Skeltved et al.*, 2014). Therefore, we think it is important to provide the researchers of the community with clear tests frameworks, together with reference data, to make it possible for them to benchmark their custom made codes. On our side, with this series of two GMD papers (*Rutjes et al.* (2016) and this one), we are settling the first foundations of a larger, long-term work.

**However, we have to understand, that M-C simulations for such a complicated domain as High-energy Physics in Atmosphere (HEPA) is not a precise tool! We don't know the distribution of the electric charges in the cloud and, therefore, strength and elongation of the emerging electric fields. Therefore, very time-consuming and detailed verification of different M-C programs, for opinion of this referee is not too important on the present stage of HEPA progress. As I mention in my review to the first paper of this series in 2016, the validation of the available experimental observations is vital for the progress of HEPA. There are published numerous gamma ray energy spectra observed on the mountain altitudes and few electron energy spectra; why not to try to compare simulations with observations? Continuous simulations with different codes and arbitrary parameters (sometimes nonrealistic, see my comment below to 4.3.1) can make illusion of intense scientific research; however only comparisons with observations and physical inference on the observed phenomena really values. Sure, authors will argue that model validation is out of scope of their paper. And they will be right. However, I can ask, when they will use their verified models for coming with comprehensive model of HEPA? When they will develop models with realistic parameters and compare it with data (energy spectra measured on Earth's surface and in the space)?**

Comparison with experimental data is not the goal of this work. However, we think it is worth mentioning here that we are working on other projects in parallel, for which having tested the coherence between these three different codes beforehand is important. Also, we would like to point out that the REAM model, tested in this study, was used to analyze experimental data detected from space and from the atmosphere in many occasions (see *Dwyer et al.* (2012)). Furthermore, compared to our status in 2016, a part of the team involved in the present study (including the first author) is now conducting, in parallel, Geant4 simulations attempts in comparison with experimental data, through two separate studies. They both concern aircraft-based measurements, one from 12 km altitude (ILDAS campaign) and the other from 20 km altitude (ALOFT/FEGS campaign); and there also having first found that

there is a form of consensus between the different available RREA models is important to interpret the results. Some of these results have been very recently published *Kochkin et al.* (2018).

**[...] Continuous simulations with different codes and arbitrary parameters (sometimes non-realistic, see my comment below to 4.3.1) [...]**

See answer below.

**To be not too didactic, I'll cite our old paper, where we try for the first time to compare simulated energy spectra with measured ones and establish a TGE model (see Figs 8-12 of Chilingarian, Mailyan and Vanyan, 2012). "With newly estimated thundercloud height, we re-estimate several phenomenological parameters of the RREA process as the following: the most probable height of thundercloud (and electrical field therein) is ~50 m. The number of electrons with energies ° above 1 MeV at the exit from the cloud is 1.97107 electrons/m2/min; if we assume that the radiation region in the thundercloud has a radius of 1 km the total number of ° electrons crossing this region in a minute is ~61013.**

**Sure, we use not optimized M-C; and, maybe we make some mistakes in our inference. We discuss possible sources of the systematic errors:**

**"We do not measure the electric field within the thundercloud; near surface electric field is not a good proxy of the intracloud fields accelerating electrons downward. We also do not measure vertical extension of the field and only estimate the height of the cloud. Therefore, simulations of the RREA process in the atmosphere with chosen parameters, although are in an agreement with the available measurements of electric fields in the thunderclouds, cannot be used for direct comparisons with TGE measurements. However, these simulations give us understanding of the RREA scale and MOS processes and expected behavior of the energy spectra."**

**However, it was the first time we present gamma ray and electron energy spectra along with simulations and achieve overall agreement. Now we develop a new method of cloud height estimation and can approach observations with more realistic simulations with more reliable better parameters. What I want to demonstrate is that simulation should be paired with experimentation; and each should profit from other.**

We thank the referee for pointing out and explaining some of the very interesting projects his team has been working on. Fortunately, his review will stay available for future use on the GMD webpage of our article.

**After our recently observation of Long Lasting Low energy TGEs (LLL TGEs) – a hours extending flux of gamma rays of 0.3 − 3 MeV energies, we started a cycle of simulations to get answer if remote Extensive Cloud Showers (ECSs) can contribute to this flux, or we should consider stochastic electron acceleration by a "sea" of randomly distributed charges in the thundercloud. Thus, simulations are pairing with observations and with hypothesis testing.**

We thank the referee for pointing out this very interesting work in progress.

**Specific questions :**

**5 The emerging field of High Energy Atmospheric Physics studies events producing high energy particles and associated with thunderstorms, such as terrestrial gamma-ray flashes and gamma-ray glows. Without mentioning Thunderstorm Ground Enhancements (TGEs) this statement is biased. The difference in duration between TGF and gamma-ray glows can be explained by two possible different scenario to create runaway electrons. Largest TGE detection prove that long duration can be explained by the continuous acceleration and multiplication of seed electrons entering strong prolonged electric field. Such a condition can sustain for minutes and, so called, extensive cloud showers (ECSs) will produce fluxes of electrons, gamma rays and neutrons on the earth's surface, i.e. TGEs (see Chilingarian et al., 2017).**

In the first version of the manuscript, the expression "Thunderstorm Ground Enhancements (TGEs)" is mentioned in the introduction (page 2, line 16 of the non-revised manuscript) but not in the abstract. We first decided not to

use it in the abstract for simplification for the reader, as we think they are not an intrinsically different phenomenon from gamma-ray glows. That is also the opinion expressed in the review paper *Dwyer et al.* (2012). However, it is not a problem to present the expression "thunderstorm ground enhancements" in the abstract too.

*page 1, line 3 of the revised manuscript : The expression "thunderstorm ground enhancements" was added in the abstract.*

In general, this issue touches on present discussions in the HEAP community, but giving a judgment of who is "right" about it, is far from the scope of our article.

**1.3 5 "The physics behind TGF, TGF afterglows and gamma ray glows are studied with the help of computer simulations, which necessarily involves model reduction and assumptions."**

**Hopefully physics is experimental science and most of results are obtained by experimentation, not simulation.**

We think that a lot of important results in the HEAP community were obtained by comparing experimental results with simulations, together with analytical calculations. The sentence has been reworded to be more adequate.

*page 7, line 4-6 of the revised manuscript : "The physics behind TGF, TGF afterglows and gamma- ray glows are studied with the help of computer simulations, which necessarily involves model reduction and assumptions."*
*→ "Apart from analytical calculations, the physics behind TGFs, TGF afterglows and gamma-ray glows are also studied with the help of experimental data, computer simulations, and often a combination of both."*

**4.3.1 5 Figure 7 compares the electron spectra recorded at 128 meters, for an electric field E = 0.80 MV/m, for a RREA generated from 200 seed electrons with ε = 100 keV. Do you especially choose the field never measured in the atmosphere (0.8 MV/m) ?**

First of all, we found in the literature one reported measurement from balloon soundings, that shows an electric field of about 200 kV/m at 12 km altitude (see *Stolzenburg and Marshall* (2009), figure 3.2 on page 65). As a reminder this is equivalent to about 0.86 MV/m at sea level, as it scales with density. In this article, such a field is associated with the balloon being close to a lightning leader. All the results are presented at STP for reference, but can be scaled to higher altitude (-> lower densities), the scaling of the different parameters being function of air density.

Furthermore, it is important keep in mind that the codes we are testing here are used by researchers that try to explore extreme hypothetical cases, because it is what may be needed to explain some gamma ray glows, and TGFs. Monte Carlo codes are routinely used by some researcher in the HEAP community in such regimes. The TGF production mechanism in particular is poorly understood, but it may require extreme potentials (>200 MV), leading to electric fields of $E = 0.80$ MV/m or more; maybe only possible for extreme thunderstorms, where measurement are not easy to get (because of rarity, and extreme conditions); maybe only possible from some extreme lightning leaders or thunderstorm configurations.

In addition, an electric field of $E = 0.80$ MV/m is about three time less the classical breakdown field $\approx 3$ MV/m at sea level, and we cannot exclude that such fields can exist, at least on a small scale and/or at high altitude (with the proper scaling factor), at which it is rather complicated to obtain in-situ measurements. In particular, simulations investigating the cold runway mechanism can require to simulate localized electric fields of 4 MV/m or more, as presented, e.g., in the recent theoretical work of *Lehtinen and Østgaard* (2018). This last article also points out that 1 MV/m is the typical field in the electrode gap used in laboratory spark experiments (typically 1 meter distance), that is also a possible application of the models used in this study; even if when simulated, these high electric fields usually extend only in a very short range (centimeter or less) and time ($\sim$ micro-second scale).

*Page 3, lines 8-13 of the revised manuscript: For completeness, a sentence was added in the introduction about x-ray emissions observed in laboratory spark experiments with a series of interesting references.*

**Or it is not in atmosphere?**

All the electric fields of this study are applied in the earth's atmosphere at sea level, Standard Temperature and Pressure (STP).

**Why 128 m? Where do you inject 200 electrons?**

For the results presented in this figure, 200 electrons are generated at the origin $(x = 0, y = 0, z = 0)$, the record is made at 128 m $(x = 0, y = 0, z = 128)$, $z$ being the direction of the electric field, and electrons are accelerated towards positive $z$.

*This was clarified throughout the text, also responding to a comment of referee #2. See page 24, figure 7; page 25 figure 8; page 22, line 19 and page 23, line 21.*

This distance of 128 m is chosen because it corresponds to a large enough number of avalanches lengths (about 8.5 avalanche lengths in this case), that corresponds to a maximum multiplication factor of about 5000, where we are sure that RREA is fully developed and has reached the self-similar state. Such multiplication factor were also necessary to be able to produce a database with enough statistic to build, with low noise, the avalanche length and avalanche times curves (figure 3 and 4).

*page 22, line 20-21 of the revised manuscript : a sentence was added to justify the record at z=128 meters.*

Spectra where the particles are recorded at sorter distances are also presented in the supplementary material, and correspond to snapshots of the RREA evolution with lower multiplication factors.

Furthermore, a multiplication factor of 1000 or more can be obtained by some groups when they attempt to model possible TGF production mechanisms (see, e.g., (*Dwyer*, 2008; *Carlson et al.*, 2010)), and may also be necessary to explain the most extreme gamma-ray glow observations, where increases of the background intensity of this order of magnitude were detected (see, e.g., (*Eack et al.*, 1996; *Eack et al.*, 2000), and some up-coming glow studies).

**One way of deciding which model is the most accurate might be to compare these results with experimental measurements. but in the context of TGF and Gamma-ray glows it is complicated to get a proper measurement of electron spectra produced by RREA. Finally, yes, only way to decide which model is true is the comparison with experiment that is missing in this paper.**

See response to the general comment above.

**Others changes (not directly suggested by referees)**

During the revision process, several extra improvements to the paper were suggested by the authors:

- *Page 16, line 26 of the revised manuscript: A citation to the article "Fundamental parameters of the relativistic runaway electrons avalanche in air" by Babich et al. (2004) was added; as it is an important study to mention in the context of this work.*

- *Page 2, line 21 of the revised manuscript: For completeness, we added two citations to the recent gamma-ray glow observations of (Kochkin et al., 2018; Dwyer et al., 2015) in the introduction.*

- *abstract (page 1-2): Improvements in the English, added more details on the Electric field range.*

- *conclusions (page 26-27): Improvements in the English, added more details on the Electric field range, and more details.*

- *The second paragraph of the introduction (page2, line 11-13) was updated to give two more interesting citations about TGF satellite observations (one for RHESSI, one for AGILE and a more recent one for Fermi).*

- *Figure 2: The 10%, 50% and 90% probability contours for the REAM model (red curves) were added, together with a paragraph discussing how it compares with Geant4 (Page 13-14, lines 34-35 and 1-9).*

- *Page 3, lines 29-32 of the revised manuscript: We added a small paragraph clarifying the differences between the different values (between 2.36 and 3.2 MV/m) of the classical breakdown field, that can been seen in the literature.*

- *Page 13, lines 4-5 of the revised manuscript: a sentence was added to justify the use of this particular $\{E, \epsilon\}$ set.*

- *Page 23, lines 1-4 of the revised manuscript:* a sentence indicating why an electric field of $E = 0.80$ MV/m is used for the comparison case was added.

- *Page 13, lines 15-16 of the revised manuscript:* an indication that we provide Geant4 examples source codes with tweakale $\alpha_R$ and $\delta\ell_{max}$ parameters was added.

- Page 12, lines 23-25 of the revised manuscript: We added a sentence about the chosen direction of the initial electrons in the RREA probability simulation and its impact on the probability.

**2 Response to the review of Referee # 2 (with relevant changes listed with italic font)**

**Response to referee #2**

We thank the Referee for his careful reading, and the valuable comments and suggestions he made that helped to improve the quality of the manuscript. We considered each of the comments and questions and provided an adequate answer. The comments of Referee #2 are written in bold and the answers in plain text. Sentences indicating a modification to the manuscript are written using italic font.

The latex diff of the revised manuscript, accounting for comments of referees #1 and #2 is provided as an attached file.

**Response to the general comment**

**Specific questions :**

**Abstract:**

**The authors mention the "effects of electric fields" (l. 5). I assume, this study is only about ambient fields and does not include self-consistent electric fields by solving the Poisson equation.**

Indeed, the study does not include self-consistent electric fields.

*The abstract has been updated to indicate the range of electric field we used to avoid the necessity to have self consistent models.*

**Also, in line 10, it would already good to mention what kind of "stepping methodology" (line 10) is meant, i.e. of particles.**

*page 1, line 10 of the revised manuscript: The sentence has been updated to indicate "[...] stepping methodology of particles [...]".*

**The authors mention that they only tested electric fields until 3 MV/m; however, the electric field for thermal runaway, i.e. for all electrons to run away irrespective of their initial energy, is approx. 26 MV/m, why not consider fields between 3 MV/m and 26 MV/m?**

We did do a range of electric fields when investigating the runaway threshold, but we were limited to classical breakdown threshold for the chosen set-up / approximations. To go above classical breakdown, lower energy physics and self consistent electric fields must be considered. And the latter is not possible for Geant4 because of the space oriented, non-synchronous particle trajectories.

*The reason why we chose not use E-fields above 3 MV/m has been has been added or clarified in three parts of paper: page 3, line 27-28; page 4, line 7-8; page 8, line 13-16*

**1. Introduction:**

**1.1: line 4: If it is about observations of high-energy phenomena, I would suggest to cite Fishman et al., 1994 in addition to Williams, 2010.**

We agree that citing (*Fishman et al.*, 1994) is relevant here.
*page 2, line 9 of the revised manuscript: the suggested citation was added.*

**1.2: In line 14, the authors talk about the energy regime of HEAP. However, they do not define this energy regime. Please be more precise in defining the energy range.**

The energy regime we are considering here is up to about 100 MeV. However we think now that it was over-stating to qualify it "the energy regime of HEAP", since, some researchers in the community may encounter higher energies, e.g. when working on cosmic rays and lightning.

*page 4, line 2 of the revised manuscript: "In the energy regime of HEAP, ..."* → *"In the energy regime of a kilo-electronvolt to a hundred of mega-electronvolts, ...".*

**Futhermore, the authors write that some individual electrons do not survive. Especially, for high-energy electron beam, it would be good to name the reason for this.**

Any electron has a chance to undergo "hard" collisions where a large amount of energy is transferred to the photon in a bremsstrahlung or ionisation interaction. In such interaction, if the energy drops below the runaway threshold (or the energy threshold, as both should ideally bet set to similar values), it is a process that removes a high energy electron from the considered population. We agree with the referee that it can be indicative to give this reason in the article.

*page 4 line 16-17 of the revised manuscript:* the reason just described here was added.

**Also, please specify what values you consider "much larger than the ionisation threshold" (page 3, lines 23-24).**

Such a value can be set to above a few keV.

*page 4, line 19-20 of the revised manuscript: an indicative value of "a few kilo-electronvolts" is now specified.*

**On page 4, lines 14-15, the authors write "The minimum energy $\epsilon_2^{\min}$ that can runaway is given by the requirement $F(\epsilon_2^{\min}) > E$ [...]." But should the friction force not be smaller for runaway? Please clarify this.**

The friction force is split into two parts. One part is below the electron low energy cutoff and used as a friction. The other part is considered by explicit discrete collisions. This first friction part for electrons above 10 keV is indeed a function of energy and decreases for increasing electron energy, but converges to a constant for electron energies above 1 MeV. So the lower the energy, the higher the friction and the threshold $F(\text{electron energy}) = qE$, lies orders of magnitude higher than ionization energy (i.e. $\approx 10$ eV comparing to $\approx 10$ keV).

Also, to be more correct, it is $q \times E$ that has the unit of the friction force $F$.

*page 5, line 13 of the revised manuscript: "$F(\epsilon_2^{\min}) > E$" was changed to "$F(\epsilon_2^{\min}) \approx q\,E$"*

**On page 5, the authors discuss the angular dependency (between the electron motion and electric field direction) on the run-away process. This has already been discussed very extensively by [O. Chanrion et al., 2016. Influence of the angular scattering of electrons on the runaway threshold in air. Plasma Phys. Contr. Fus., vol. 58, 044001]. Please cite this article.**

This article was already cited in another section of the non-revised manuscript (p.2, line 28). But we agree that citing it again inside this section discussing the angular effects is a good idea.

*page 6, line 22 of the revised manuscript: A second citation to the article (Chanrion et al., 2016) was added.*

**2. Model descriptions:**

**2.1: The authors say that different sets of electro-magnetic cross sections are used. However, the authors do not state (neither in the main text nor in the supplementary material) which processes are actually taken into account. This is clearly missing, but crucial since simulation results strongly depend on the chosen processes and cross sections. The authors state where the cross sections come from and they make some comparison plots in section 8 of the supplementary material, but only for a few processes. Say, in the future, other researchers want to compare their results. Then, the knowledge of the used cross section data is crucial to interpret results. I would thus suggest to elaborate more on the processes and cross sections.**

We agree with this point: it would be more practical for future researchers that would like to use this work to have an easy-to-read summary of the process and which cross section sets are used by the models we tested.

*We added a table providing all this information (i.e. processes cross sections and/or models used by each code) as part of the supplementary material .pdf file (section 13). We also added a reference to this table in the revised manuscript (page 8, lines 4-5).*

**2.2: On page 8/line 29, the authors say that GRRR uses the "energy at that instant" to calculate the collision rate $\nu_k$. However, it is not clear which energy: the energy of each individual particle, the maximum energy of all particles or the mean energy of all particles. Please clarify.**

Thanks for pointing out this ambiguity. Now we state clearly that it is the energy of each particle that is used to calculate collision rates.

*page 10, line 9 of the revised manuscript: "the rate $\nu_k$ is calculated using the energy at that instant" → "the rate $\nu_k$ for each particle is calculated using that particle's energy at that instant".*

**2.4.1: For space-oriented codes, "a single particle is simulated over its entire life-time" (page 9/line 21). However, what is the reason to lose an electron (especially in the regime above several keV). The only reason to lose an electron would be attachment to air molecules. However, for this process to occur the electron normally needs to lose more energy than down to 10 keV. Please be more precise here.**

Our wording "a single particle is simulated over its entire life-time" was not accurate, since the particles are usually simulated until they go below the energy threshold, for most of the codes. Note that Geant4, by default, does simulate all primary particles down to zero energy (the "energy threshold", in Geant4, is a minimum energy for the secondary produced particles to be tracked or not). As shown in the section 3 (RREA probability study), the probability for an initial electron to produce a RREA is negligible below 10 keV. In this energy range, the electrons will slow down in a small length and time scale; and indeed it is only when they go down to energy in the eV scale that they will attach to air molecules and that we can consider that it is "really lost"; and this will happen a little bit later in their lifetime.

*Page 1, line 4 of the revised manuscript : "a single particle is simulated over its entire life-time" → "is simulated until it goes below the low energy threshold $\epsilon_c$, chosen by the user". We also repeated the precison that Geant4 follows all primary particles until zero energy (that was also indicated before in the Geant4 description section 2.1).*

**2.4.2: Why would "acceptable values of $\delta\ell_{max}$ depend on the electric field" (page 10/line 28)? It is very clear that it should be smaller than the electron's mean free path. But the mean free path depends on the electron energy rather than on the electric field. Please clarify the dependence on the electric field.**

The gain or loss in energy by a particle due to the E-field depends on the distance it propagated along its direction, and its magnitude. The electron may move from point A to point B, and could do a collision or a null-collision at point B. The result of the collision at point B depends on the (differential) cross sections at this point, that depends on the energy, that depends on the energy it gained (or lost) when traveling from A to B, that depends on the electric field and the distance between the two points , that will be affected by $\delta\ell_{max}$ in some case. For the tracking to be accurate, the cross sections should remain approximately constant between point A and B, that is

only acceptable if the A-B distance is not too high. The mean free path is important, but so is also the distance at which the gain in energy due to the electric field is small enough so that the change in the cross sections between point A and B is small enough. Depending on the context, it could be greater or smaller than the mean free path, and one should always use a step length that is small enough compared to both quantities. In Geant4, that can be achieved by forcing the maximum acceptable step $\delta\ell_{max}$ to be small enough, or, more cleverly, by reducing the $\alpha_R$ parameter. However, an exact quantification of the dependence of the good values of $\delta\ell_{max}$ (or $\alpha_R$) for different electric fields is not easy to perform, and would require extra simulations, that we did not have time to run.

However we think this explanation should not be included in the main text, since is bit too long and we could not precisely quantify the dependency of $\delta\ell_{max}$ with $E$. Thankfully, this response to the reviewer will be available in the GMD web-page of this paper.

**3. Probability of generating RREA:**

**In Figure 1, the authors present the probability that a single electron with an initial energy of 75 keV in an ambient field of 0.8 MV/m creates an RREA. As a criterion, they use that at least 20 electrons with an energy of 1 MeV are created. Please specify which value for $\delta\ell_{max}$ is used in panel a) and which value for $\alpha_R$ is used in panel b).**

The values are the default ones. That is $\delta\ell_{max} = 1$ km and $\alpha_R = 0.2$ for O4 and $\alpha_R = 0.8$ for O1.

*page 14 of the revised manuscript: The caption of figure 1 has been updated to indicate this.*

**However, I am confused, though. The friction force based on [A.V. Gurevich, 1961. On the theory of runaway electrons. Sov. Phys. JETP-USSR, vol. 12, pp. 904–912] is supposed to be 0.65 MeV/m for a 75 keV electron, thus the ambient field is definitely sufficient to accelerate the electron into the run-away regime. Of course, this does not mean that 20 electrons with energies are above 1 MeV, but 12% seems low.**

The RREA probability value for this case is around 10 to 12% and is given by two independent codes (GRRR and Geant4). We think that going from the (average) friction force to this probability is by no means straightforward, as it is a stochastic process, and it requires computer simulations to be evaluated (as far as we known). We encourage other researchers to evaluate this probability independently, and maybe by different methods from Monte-Carlo if possible.

**But this might depend on the simulation time. How long has the shower been simulated?**

The simulation is running for as much time as needed for one of the two possibilities to happen for each single electron seed:

- [i]: it is absorbed

- [ii]: there is at least 20 electrons of more than 1 MeV in the system.

The simulation space is large enough so that the particles will never encounter the border. In pratice, for this configuration ($\epsilon = 75$ keV and $E = 0.80$ MV/m), the electron time is $\approx 2$ ns (times are given from the global / laboratory fame) if [i] is reached and $\approx 230$ ns on average if [ii] is reached. For information, it takes $\approx 190$ ns on average if 10 electrons of more than 1 MeV are required, $\approx 130$ ns for 5, and $\approx 17$ ns for 1.

**It would also be good to see the RREA percentage for different criteria (20 electrons above 1 MeV, 10 electrons above 1 MeV, 5 electrons and finally 1 electron above 1 MeV which should give almost 100%).**

The RREA percentages as function of the number of 1 MeV electrons requested are presented here for two cases: ($\epsilon = 75$ keV, $E = 0.8$ MV/m) and ($\epsilon = 800$ keV, $E = 0.35$ MV/m). It was obtained using the Geant4 O4 model with $\alpha_R = 0.001$.

[Figure]

For the ($\epsilon = 75\,\text{keV}$, $E = 0.8\,\text{MV/m}$) case, the probability does not significantly changes if 20, 10, 5 or 1 electron of more than 1 MeV are required. This result is actually consistent with the RREA probabilities we found for higher energy electron seed with the 0.8 MV/m field (see supplementary material section 5.2, or the figure just below here). Once the seed electron has reached 1 MeV, then its probability to generate the 20 MeV electrons in the 0.8 MV/m field is 100 %. Actually, for this values of ($\epsilon, E$), what mainly affects the final probability is what happens at the start of the simulation. i.e. if the single initial electron is able to gain enough energy, or goes below the energy threshold before (that is here set to 990 eV). However, for ($\epsilon = 75\,\text{keV}$, $E = 0.8\,\text{MV/m}$), the "required number of 1 MeV electrons" does play a significant role in the probability. It is coherent with what is indicated in the manuscript: "The difference [between requiring 1 or 20 MeV electrons] is mainly noticeable for low electric field ($< 0.4$ MV/m) and high seed energies ($> 700$ keV)." (page X, lines XX)

*Supplementary material, section 5.3: the previous plot was added as the reader may be interested to know this information.*

*page 12, line 32-33 of the revised manuscript: a reference to this plot in the supplementary material was added.*

**In Figure 2, the authors show the avalanche probabilities (10%, 50% and 90%) as a function of initial electron energy and ambient electric field. What about the right top (high energy, high field) and bottom left (low energy, low field) part? What are the probabilities there?**

Plots indicating the values for the other probabilities for the four models are presented in the supplementary material (section 5). For high field or high energy, the probability is 100%, For low field and low energy, the probability is 0%.

However, the primary objective of figure 2 was not to present the full probability domain, but to compare the codes for three probabilities in-between the 0% and the 100% case, where we expected to see most of the differences. Furthermore the 0% and 100% contour lines tend to be more noisy than the three other ones, making it more difficult to compare the models. However for the Geant4 O4 model (supposingly the most accurate model), we could build a distribution with very large statistics ($>$50,000 seeds per parameter set), and we could produce a noiseless plot with the 0%, 10%, 50%, 90% and 100% probabilities contour lines :

[Figure]

**There is so much space in this figure. Why not add some curves or values for these two regimes.**

*As suggested by thre referee, we think it will be helpful for the reader to see the level curves corresponding these two regimes, i.e. the 0% and the 100 % levels. So we integrated this last plot in the manuscript in addition to Figure 2 (→"Figure 2.b."). And the former "Figure 2" is now "Figure 2.a."*

However we don't want to add more level curves (e.g. from other codes) to Figure 2.a. because it will make it hard to read.

**4. Characterisation of RREA showers:**

**When discussing the evolution of the self-similar state , the authors say that they used a different number of seed electrons for Geant4, REAM and GRRR (page 16, lines 12–13). For consistency, I suggest to add one more case where the same number of seed electrons is used.**

For very large electric field ($> 2$ MV/m), Geant4 used 200 seed electrons, but GRRR only used 20 and REAM used 16, due to limitations in our computation resources. In addition for low E-fields, only a few of the 200 sampled electrons did trigger a RREA for O1 and REAM (as the RREA probability is low for $\epsilon = 100$ keV, $E = 0.8$ MV/m), meaning also larger error bars for the two models. As we do not have the capability to run GRRR or REAM with larger statistics, the cases with the same number of seed electrons would mean showing Geant4 results with lower statistics. But we think this is not really necessary: it will make the figure less readable, without providing extra information. Actually, the uncertainty intervals for Geant4 simulations with reduced statistics can be easily guessed: they would expand by about a factor of 3.2 ($\approx \sqrt{\frac{200}{20}}$) for 20 seeds and 3.5 ($\approx \sqrt{\frac{200}{16}}$) for 16 seeds.

**Figure 5 shows the time to reach the self-similar state. Comparing all the different models, it seems that the time to reach that is consistent within one order of magnitude. I propose to add this to line 14 on page 16.**

Indicating that the models are consistent within one order of magnitude is largely over-evaluating the difference, since:

- Without considering the error bars, the relative difference between the models is never more than 200%

- for most of the electric fields, the models are consistent within the statistical error bars; that are a bit large for REAM and GRRR due to the limited amount of seed electrons we could simulate

We think the way it is is already discussed in the non-revised manuscript is good and should not be changed.

**4.3.1: In Figure 6, the authors present the mean energy (of the self-similar state) as a function of the ambient field. It might be good to remind the reader in the figure caption which value for $\alpha_R$ was used here for O1 and O4.**

The value of $\alpha_R$ is set to the default for O1 and O4, that is 0.8 and 0.2 respectively. In this case the maximum acceptable step of 1 millimeter ensures that we have accurate simulations.

*Page 21: the caption of figure 6 has been updated to give the suggested information.*

**I am wondering how the plots for O1 and O4 would look for $\alpha_R$. It would be good to plot one case for a different $\alpha_R$ to show the dependence of the mean electron energy as a function of $\alpha_R$ .**

The main point of this section (4.3) of the paper is to compare the models in the case were we think they are set-up optimally, so we do not think that adding this $\alpha_R$ comparison in this section is a good idea. In the original (and revised) manuscript, we just show one case where the maximum acceptable step is too large to give the reader an idea of how bad it can be, without discussing it further.

However it is informative to see if reducing $\alpha_R$ makes the mean energies converge towards similar values compared to reducing $\delta\ell_{max}$ to 1 mm. *We added plots of mean energy as function of $\alpha_R$ for O1 ($\delta\ell_{max}$ is set to 1 cm to have a common reference with figure 6), in the supplementary material (section 7), for several electric fields.* Unfortunately, we could not cover the same number of electric fields, and could not simulate the same number of seed (i.e. the new data is more noisy).

*Page 21, line 12-13 of the revised manuscript: A reference to the supplementary material pointing to this curve was added.* Here is a reproduction of this curve:

[Figure]

As expected, $\delta\ell_{max} = 1$ mm is close to $\delta\ell_{max} = 1$ cm with $\alpha_R = 0.001$

**In Table 2, the authors present the fit parameters $a_1$ , $a_2$ and $a_2$ of Eq. (20) for different models. Please add the error bars in order to judge the quality of these fits.**

*Table 2 (page 22) has been updated to give confidence intervals on the fitted parameters $a_1$ , $a_2$ and $a_2$, and the text was also updated accordingly (page XX, line XX-XX). The best fit values have also been re-evaluated compared to the previous version of the manuscript.*

We want to make clear that the confidence intervals given are not error bars. They indicate what is the range of values of the three parameter that are able to give good quality fits, assuming a 95% confidence threshold. The quality of the fits can be evaluated using the $r$-squared coefficient, and it is larger than 0.90 for all the fits of table 2, meaning that the model (equation 20) fits very well the data, as can be qualitatively observed in Figure 6.

**In Figure 7, the authors present the electron spectra at 128 m. But does 128 m refer to the z-coordinate or to the travelled distance $r = (x^2 + y^2 + z^2)^{1/2}$. Please clarify this in the figure caption and in line 5 (page 19), line 2 (page 20) and in the caption of Figure 8.**

The distance of 128 meters refers to the Z coordinates (-Z being the direction of the E-field so that electrons are accelerated towards positive Z).

*page 24, figure 7, page 25 figure 8, page 22, line 19 and page 23, line 21: the manuscript were updated as requested.*

**4.3.2: In section 4.3.2, the authors state that a comparison with photon measurements is difficult because of the attenuation of photons in air. Whereas I agree in general, there are some issues I would like to address. The authors say that a 100 keV photon at 12 km altitude travels 1540 m in average, a 50 keV photon 671 m and a 20 keV photon 63 m. Where do these values come from? Is this a result of their simulations (if so, how did you obtain these); if not, please cite your source. I made a brief comparison with NIST data (http://physics.nist.gov/PhysRefData/Xcom/Text/XCOM.html) and obtained attenuation lengths of approx. 2000 m for 100 keV, 1600 m for 50 keV and 500 m for 20 keV. This needs to be clarified.**

The given average path lengths come from Geant4 simulations. It is a simple set up where $M$ >10,000 mono-energetic photons are sampled (in air at 12 km density) and their total path length before absorption is recorded, and then the $M$ length values are averaged. The standard deviations around the averages are also calculated and given in the article. If requested, we would be glad share the Geant4 source code.

We agree with the attenuation lengths given by the referee. However, the attenuation length and the average path length are different quantities: the average path length is lower because a photon will not keep the same attenuation length over his full track (it will be reduced at each step) since it is losing energy along its path. These mean path lengths values also have large variations from event to event (stochastic process) that are evaluated with the standard deviations indicated in the manuscript.

*Page 24, line 2-4 of the revised manuscript : to clarify this point, a sentence was added.*

**Additionally, the authors say that the "photons have no chance to escape the atmosphere and to be detected by a satellite" (pages 20/21). I would like to remind the authors that the "average path travelled" (page 20/line 14) or the attenuation length is only an average. Hence, there can always be photons which escape the atmosphere and may be detected at satellite altitudes even though the probability is low. Actually, Fermi has measured photons with energies between 10 and 500 keV (see https://gammaray.nsstc.nasa.gov/gbm/science/terr_grf.html). Please be more precise here.**

It is close to impossible for a photon of less than 500 keV to reach space, if it is emitted from 12 km altitude (and even more impossible for 100 keV photons). The vast majority of photons (if not all of them) observed below 500 keV by the Fermi space telescope (and also RHESSI) are actually photons that, at emission, had larger energies (likely more than 1 MeV) and lost some part of it by collisions with air, and eventually reached the satellite.

*Page 24, line 6-11 of the revised manuscript : to clarify this point in the paper, three sentences was added.*

For information, here is a plot giving the probability of photons to escape the atmosphere as function of energy, assuming a source at 12 km altitude:

[Figure]

Note that, even for 40 MeV photons, the probability to escape is less than 1%, indicating that if a TGF is produced from this altitude, it must be very strong to be detected in space. The curve was obtained from Geant4 simulations and, if requested, we would be happy to share the source code.

*Supplementary material, section 14 : the previous curve was added, as it may be helpful for other researchers.*

**4.4: In the supplementary material, section 9.2., I cannot find any plot showing the parallel velocity $\beta_\parallel$ (only the mean Z speed; or is the mean Z speed meant to be $\beta_\parallel$ ). Maybe, it is there, but at least not apparent. Could the authors please point me to the correct plot?**

The mean $\beta_\parallel$ and the mean Z speed are indeed the same, since the the electric field is on the Z direction. We only compare mean $\beta_\parallel$ values and do not show the full distributions of $\beta_\parallel$.

*Page 25, line 4 of the revised manuscript : the sentence has been updated to clarify this.*

**page 24, lines 24/25: The authors say that single Coulomb scattering would "increase the necessary computation time". Should using a single scattering (instead of multiple scattering) not decrease the computation time?**

The number of interactions experienced by an electron (or positron) before being stopped increases with its kinetic energy and so a detailed simulation becomes very demanding in computation time at high energies: that's why multiple scattering method were developed. The idea behind the multiple scattering algorithms is to avoid to explicitly simulate every "hard" collision of every single electron (i.e. avoid doing single scatterings), but to do multiple scatterings inside one step length (or one collision). It permits to use step lengths substantially larger (usually >10 times) compared to a pure single scattering strategy, and so reduces the necessary computation time to simulate the propagation of a given electron (or positron), but can be less accurate. Such algorithms usually rely on several tweakable parameters that should be optimized to shorten necessary computation time while keeping an acceptable physical accuracy. Remark also that what is maybe the biggest difference between {REAM-GRRR} and {Geant4 O1-O4} (in the set-ups used in this paper), is that the former use single (Coulomb) scattering algorithms and the later uses multiple scattering.

*Page 28, lines 4-7 of the revised manuscript : the point about single scattering has been updated to clarify this.*

**Technical corrections:**

All the suggested technical correction have been applied. We thank the referee for his very careful reading.

**Others changes (not suggested by referees)**

During the revision process, several extra improvements to the paper were suggested by the authors:

- *Page 16, line 26 of the revised manuscript: A citation to the article "Fundamental parameters of the relativistic runaway electrons avalanche in air" by Babich et al. (2004) was added; as it is an important study to mention in the context of this work.*

- *Page 3, lines 8-13 of the revised manuscript: A sentence was added in the introduction about x-ray emissions observed in laboratory spark experiments, that is also a potential case where the models we are comparing can be applied.*

- *Page 2, line 21 of the revised manuscript: For completeness, we added two citations to the recent gamma-ray glow observations of (Kochkin et al., 2018; Dwyer et al., 2015) in the introduction.*

- *abstract (page 1-2): Improvements in the English, added more details on the Electric field range.*

- *conclusions (page 26-27): Improvements in the English, added more details on the Electric field range, and more details.*

- *The second paragraph of the introduction (page2, line 11-13) was updated to give two more interesting citations about TGF satellite observations (one for RHESSI, one for AGILE and a more recent one for Fermi).*

- *Figure 2: The 10%, 50% and 90% probability contours for the REAM model (red curves) were added, together with a paragraph discussing how it compares with Geant4 (Page 13-14, lines 34-35 and 1-9).*

- *Page 3, lines 29-32 of the revised manuscript: We added a small paragraph clarifying the differences between the different values (between 2.36 and 3.2 MV/m) of the classical breakdown field, that can been seen in the literature.*

- *Page 13, lines 4-5 of the revised manuscript: a sentence was added to justify the use of this particular {E,$\epsilon$} set.*

- *Page 23, lines 1-4 of the revised manuscript:* a sentence indicating why an electric field of $E = 0.80$ MV/m is used for the comparison case was added.

- *Page 13, lines 15-16 of the revised manuscript:* an indication that we provide Geant4 examples source codes with tweakale $\alpha_R$ and $\delta\ell_{max}$ parameters was added.

- Page 12, lines 23-25 of the revised manuscript: We added a sentence about the chosen direction of the initial electrons in the RREA probability simulation and its impact on the probability.

In this case, "all electrons" can mean only the initial (seed) electron (if it gets absorbed without being able to generate more electrons above the threshold), or several electrons that have been generated (above the threshold) by collision of the seed (and also potential secondaries of secondaries, and so on). All the potential secondary electrons also have to be absorbed for the simulation to end with condition (i).

**Could it not happen that some simulations run forever? What if there are, say, five electrons above 1 MeV running away (1 MeV in a field of 800 kV/m is well above the friction force). They might never get absorbed, but not necessarily produce new electrons above 1 MeV. Why can this option be excluded? I am not saying that this scenario might actually happen, but I cannot exclude it, either. I would be very grateful if the authors could briefly elaborate on this.**

This scenario of electrons running away forever could actually be observed in a very small amount of simulations (less than 0.1 % of all the sampled electrons), for very specific parameter sets. Therefore we did have to set up a computation time limit (corresponding to a very large number of avalanche lengths), that if reached, the electron is

discarded and not taken into account in the computation of the RREA probability. This is a technical detail that we think is not worth mentioning in the article.

**6. In the reply to referee number one, the authors say "that we do not intend to validate/verify any code in this article." This is a very interesting statement. If the authors only intend to compare the codes/models described in the paper and give some suggestions for comparisons, this is, of course, fine. However, I strongly suggest that the authors make clear that this article is not about validation/verification. That could be done in the abstract or in section 6.**

We think it is not clear what one ("one" meaning referee #1, or referee #2, or one of the authors of this article) means by "verifying". When we wrote the answer of referee #1, it is in the context of answering his comment, and he seems to indicate that verification and validation are quite similar (that can be justified true in some point of view).In this study, we do verify (or check) if different codes produce similar results in a given physical context. But we do not verify most of what can be verified, e.g. if one code is "right" or "wrong", or if they do not have mistakes in their source code, or in the calculations done for implementation the different processes, and so on.

**4    Manuscript with highlighted changes after review (marked-up manuscript version)**

[revised manuscript text omitted]